# *cis*TEM, user-friendly software for single-particle image processing

**Timothy Grant\*, Alexis Rohou†\*, Nikolaus Grigorieff\***

Janelia Research Campus, Howard Hughes Medical Institute, Ashburn, United States

**Abstract** We have developed new open-source software called *cis*TEM (computational imaging system for transmission electron microscopy) for the processing of data for high-resolution electron cryo-microscopy and single-particle averaging. *cis*TEM features a graphical user interface that is used to submit jobs, monitor their progress, and display results. It implements a full processing pipeline including movie processing, image defocus determination, automatic particle picking, 2D classification, ab-initio 3D map generation from random parameters, 3D classification, and high-resolution refinement and reconstruction. Some of these steps implement newly-developed algorithms; others were adapted from previously published algorithms. The software is optimized to enable processing of typical datasets (2000 micrographs, 200 k – 300 k particles) on a high-end, CPU-based workstation in half a day or less, comparable to GPU-accelerated processing. Jobs can also be scheduled on large computer clusters using flexible run profiles that can be adapted for most computing environments. *cis*TEM is available for download from cistem.org.

DOI: https://doi.org/10.7554/eLife.35383.001

**\*For correspondence:**
tim@tgrant.co.uk (TG);
a.rohou@gmail.com (AR);
niko@grigorieff.org (NG)

**Present address:** †Department of Structural Biology, Genentech, South San Francisco, United States

## Introduction

The three-dimensional (3D) visualization of biological macromolecules and their assemblies by single-particle electron cryo-microscopy (cryo-EM) has become a prominent approach in the study of molecular mechanisms (*Cheng et al., 2015*; *Subramaniam et al., 2016*). Recent advances have been primarily due to the introduction of direct electron detectors (*McMullan et al., 2016*). With the improved data quality, there is increasing demand for advanced computational algorithms to extract signal from the noisy image data and reconstruct 3D density maps from them at the highest possible resolution. The promise of near-atomic resolution (3–4 Å), where densities can be interpreted reliably with atomic models, has been realized by many software tools and suites (*Frank et al., 1996*; *Hohn et al., 2007*; *Lyumkis et al., 2013*; *Punjani et al., 2017*; *Scheres, 2012*; *Tang et al., 2007*; *van Heel et al., 1996*). Many of these tools implement a standard set of image processing steps that are now routinely performed in a single particle project. These typically include movie frame alignment, contrast transfer function (CTF) determination, particle picking, two-dimensional (2D) classification, 3D reconstruction, refinement and classification, and sharpening of the final reconstructions.

We have written new software called *cis*TEM to implement a complete image processing pipeline for single-particle cryo-EM, including all these steps, accessible through an easy-to-use graphical user interface (GUI). Some of these steps implement newly-developed algorithms described below; others were adapted from previously published algorithms. *cis*TEM consists of a set of compiled programs and tools, as well as a wxWidgets-based GUI. The GUI launches programs and controls them by sending specific commands and receiving results via TCP/IP sockets. Each program can also be run manually, in which case it solicits user input on the command line. The design of *cis*TEM, therefore, allows users who would like to have more control over the different processing steps to design their own procedures outside the GUI. To adopt this new architecture, a number of previously

existing Fortran-based programs were rewritten in C++, including Unblur and Summovie (*Grant and Grigorieff, 2015b*), mag_distortion_correct (*Grant and Grigorieff, 2015a*), CTFFIND4 (*Rohou and Grigorieff, 2015*), and Frealign (*Lyumkis et al., 2013*). Additionally, algorithms described previously were added for particle picking (*Sigworth, 2004*), 2D classification (*Scheres et al., 2005*) and ab-ini-tio 3D reconstruction (*Grigorieff, 2016*), sometimes with modifications to optimize their perfor-mance. *cis*TEM is open-source and distributed under the Janelia Research Campus Software License (http://license.janelia.org/license/janelia_license_1_2.html).

*cis*TEM currently does not support computation on graphical processing units (GPUs). Bench-marking of a hotspot identified in the global orientational search to determine particle alignment parameters showed that an NVIDIA K40 GPU performs approximately as well as 16 Xeon E5-2687W CPU cores after the code was carefully optimized for the respective hardware in both cases. Since CPU code is more easily maintained and more generally compatible with existing computer hard-ware, the potential benefit of GPU-adapted code is primarily the lower cost of a high-end GPU com-pared with a high-end CPU. We chose to focus on optimizing our code for CPUs.

## Results

### Movie alignment and CTF determination

Movie alignment and CTF determination are based on published algorithms previously implemented in Unblur and Summovie (*Grant and Grigorieff, 2015b*), and CTFFIND4 (*Rohou and Grigorieff, 2015*), respectively, and these are therefore only briefly described here. Unblur determines the translations of individual movie frames necessary to bring features (particles) visible in the frames into register. Each frame is aligned against a sum of all other frames that is iteratively updated until there is no further change in the translations. The trajectories along the x- and y-axes are smoothed using a Savitzky–Golay filter to reduce the possibility of spurious translations. Summovie uses the translations to calculate a final frame average with optional exposure filtering to take into account radiation damage of protein and maximize its signal in the final average. *cis*TEM combines the func-tionality of Unblur and Summovie into a single panel and exposes all relevant parameters to the user (*Figure 1*). Both programs were originally written in Fortran and have been rewritten entirely in C++.

CTFFIND4 fits a calculated two-dimensional CTF to Thon rings (*Thon, 1966*) visible in the power spectrum calculated from either images or movies. The fitted parameters include astigmatism and, optionally, phase shifts generated by phase plates. When computed from movies, the Thon rings are often more clearly visible compared to Thon rings calculated from images (*Figure 2*; [*Bartesaghi et al., 2014*]). When selecting movies as inputs, the user can specify how many frames should be summed to calculate power spectra. An optimal value to amplify Thon rings would be to sum the number of frames that correspond to an exposure of about four electrons/Å$^2$ (*McMullan et al., 2015*).

Since our original description of the CTFFFIND4 algorithm (*Rohou and Grigorieff, 2015*), several significant changes were introduced. (1) The initial exhaustive search over defocus values can now be performed using a one-dimensional version of the CTF (i.e. with only two parameters: defocus and phase shift) against a radial average of the amplitude spectrum. This search is much faster than the equivalent search over the 2D CTF parameters (i.e., four parameters: two for defocus, one for astigmatism angle and one for phase shift) and can be expected to perform well except in cases of very large astigmatism (*Zhang, 2016*). Once an initial estimate of the defocus parameter has been obtained, it is refined by a conjugate gradient minimizer against the 2D amplitude spectrum, as done previously. In *cis*TEM, the default behavior is to perform the initial search over the 1D ampli-tude spectrum, but the user can revert to previous behavior by setting a flag in the 'Expert Options' of the 'Find CTF' Action panel. (2) If the input micrograph's pixel size is smaller than 1.4 Å, the resampling and clipping of its 2D amplitude spectrum will be adjusted so as to give a final spectrum for fitting with an edge corresponding to 1/2.8 Å$^{-1}$, to avoid all of the Thon rings being located near the origin of the spectrum, where they can be very poorly sampled. (3) The computation of the quality of fit ($CC_{fit}$ - 1 in [*Rohou and Grigorieff, 2015*]) is now computed over a moving window, similar to (*Sheth et al., 2015*), rather than at intervals delimited by nodes in the CTF. (4) Following background subtraction as described in *Mindell and Grigorieff (2003)*, a radial, sine-edged mask is applied to the spectrum, and this masked version is used during search and refinement of defocus,

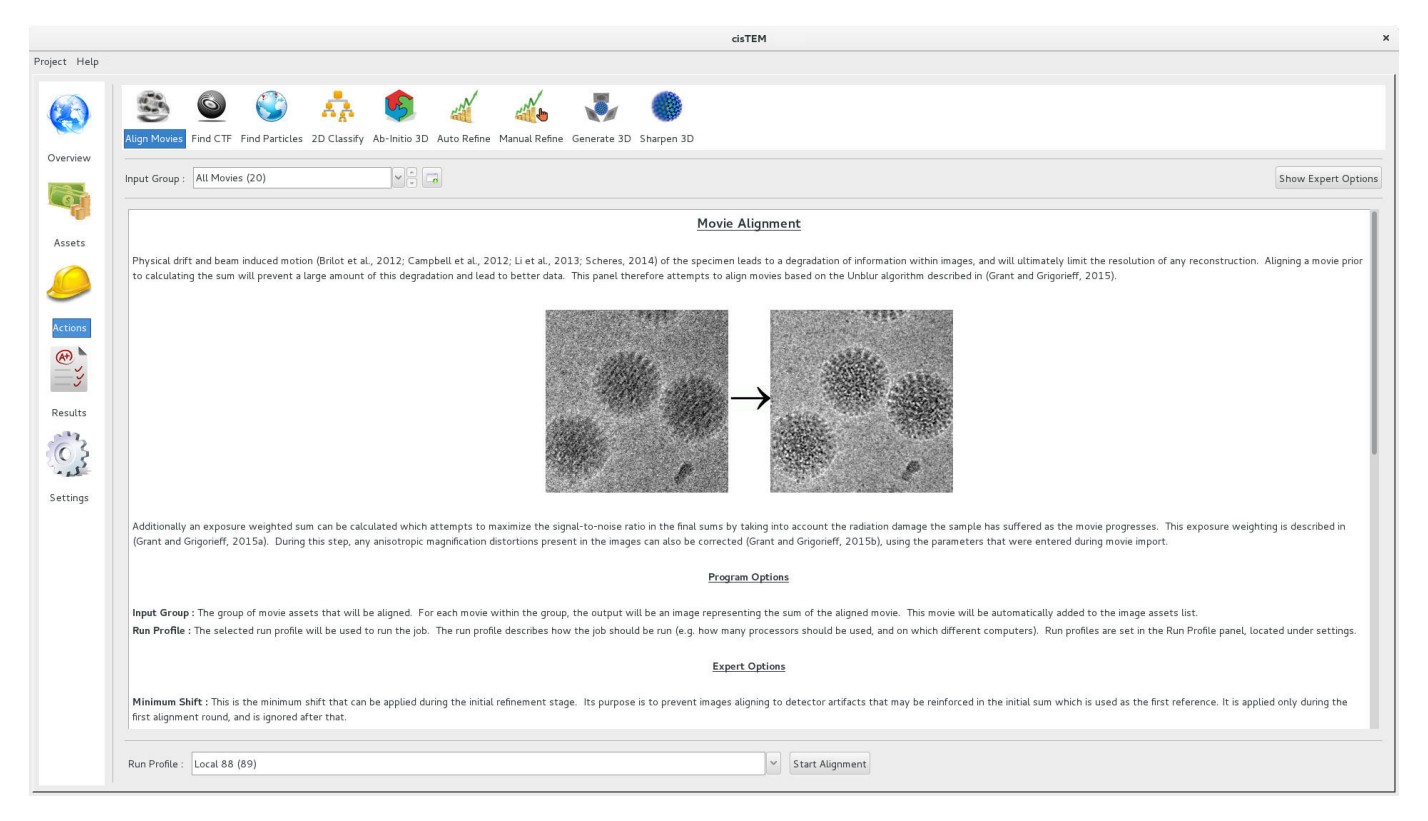

**Figure 1.** Movie alignment panel of the *cis*TEM GUI. All Action panels provide background information on the operation they control, as well as a section with detailed explanations of all user-accessible parameters. All Action panels also have an Expert Options section that exposes additional parameters.

DOI: https://doi.org/10.7554/eLife.35383.002

astigmatism and phase shift parameters. The sine is 0.0 at the Fourier space origin, and 1.0 at a radius corresponding to 1/4 Å$^{-1}$, and serves to emphasize high-resolution Thon rings, which are less susceptible to artefacts caused by imperfect background subtraction. For all outputs from the program (diagnostic image of the amplitude spectrum, 1D plots, etc.), the background-subtracted, but non-masked, version of the amplitude spectrum is used. (5) Users receive a warning if the box size of the amplitude spectrum and the estimated defocus parameters suggest that significant CTF aliasing occurred (*Penczek et al., 2014*).

## Particle picking

Putative particles are found by matching to a soft-edged disk template, which is related to a convolution with Gaussians (*Voss et al., 2009*) but uses additional statistics based on an algorithm originally described by *Sigworth (2004)*. The use of a soft-edged disk template as opposed to structured templates has two main advantages. It greatly speeds up calculation, enabling picking in 'real time', and alleviates the problem of templates biasing the result of all subsequent processing towards those templates (*Henderson, 2013*; *Subramaniam, 2013*; *van Heel, 2013*). Any bias that is introduced will be towards a featureless 'blob' and will likely be obvious if present.

Rather than fully describing the original algorithm by (*Sigworth, 2004*), we will emphasize here where we deviated from it. The user must specify three parameters: the radius of the template disk, the maximum radius of the particle, which sets the minimum distance between picks, and the detection threshold value, given as a number of standard deviations of the (Gaussian) distribution of scores expected if no particles were present in the input micrograph. Values of 1.0 to 6.0 for this threshold generally give acceptable results. All other parameters mentioned below can usually remain set to their default values.

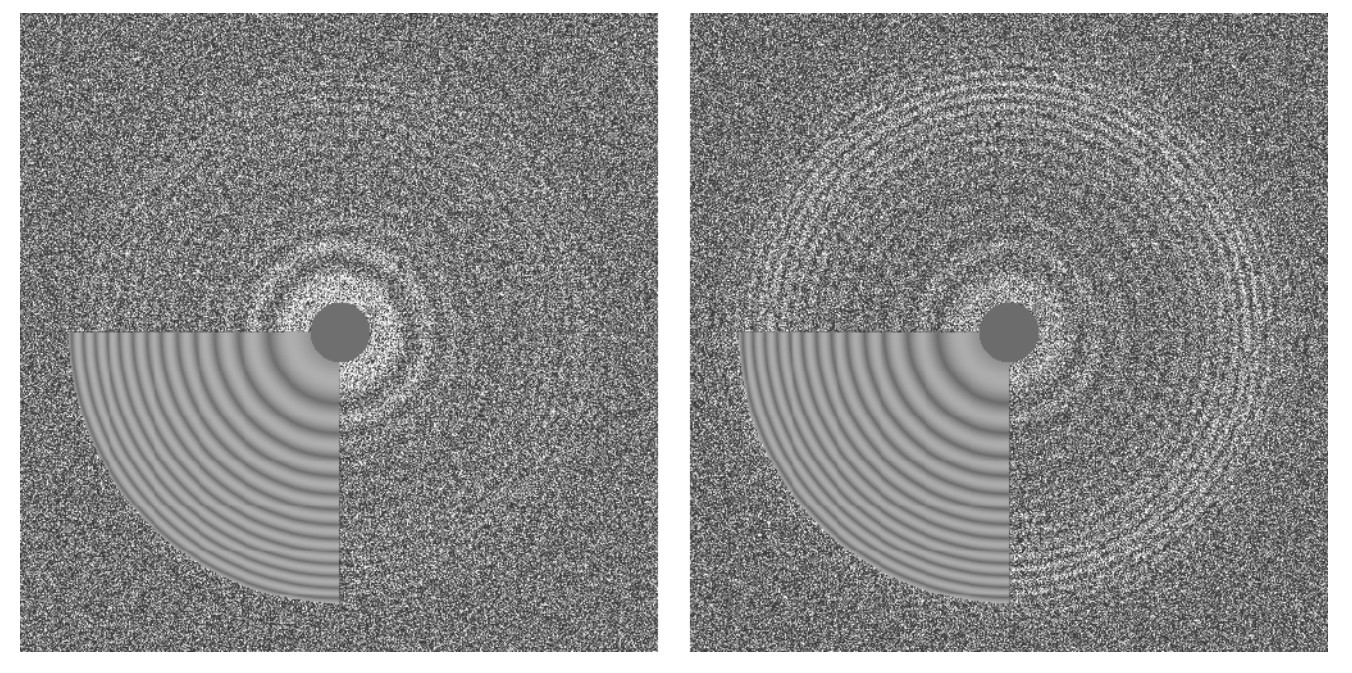

**Figure 2.** Thon ring pattern calculated for micrograph '0000' of the high-resolution dataset of β-galactosidase (*Bartesaghi et al., 2015*) used to benchmark *cis*TEM. The left pattern was calculated from the average of non-exposure filtered and aligned frames while the right pattern was calculated using the original movie with 3-frame sub-averages. The pattern calculated using the movie shows significantly stronger rings compared to the other pattern.

DOI: https://doi.org/10.7554/eLife.35383.003

Prior to matched filtering, micrographs are resampled by Fourier cropping to a pixel size of 15 Å (the user can override this by changing the 'Highest resolution used in picking' value from its default 30 Å), and then filtered with a high-pass cosine-edged aperture to remove very low-frequency density ramps caused by variations in ice thickness or uneven illumination.

The background noise spectrum of the micrograph is estimated by computing the average rotational power spectrum of 50 areas devoid of particles, and is then used to 'whiten' the background (shot +solvent) noise of the micrograph. Normalization, including CTF effects, and matched filtering are then performed as described (*Sigworth, 2004*), except using a single reference image and no principal components' decomposition. When particles are very densely packed on micrographs, this approach can significantly over-estimate the background noise power so that users may find they have to use lower thresholds for picking. It might also be expected that under those circumstances, micrographs with much lower particle density will suffer from a higher rate of false-positive picks.

One difficulty in estimating the background noise spectrum of the micrograph is to locate areas devoid of particles without a priori knowledge of their locations. Our algorithm first computes a map of the local variance and local mean in the micrograph (computed over the area defined by the maximum radius given by the user [*Roseman, 2004*; *Van Heel, 1982*]) and the distribution of values of these mean and variance maps. The average radial power spectrum of the 50 areas of the micrograph with the lowest local variance is then used as an estimate of the background noise spectrum. Optionally, the user can set a different number of areas to be used for this estimate (for example if the density of particles is very high or very low) or use areas with local variances closest to the mode of the distribution of variances, which may also be expected to be devoid of particles.

Matched-filter methods are susceptible to picking high-contrast features such as contaminating ice crystals or carbon films. (*Sigworth, 2004*) suggests subtracting matched references from the extracted boxes and examining the remainder in order to discriminate between real particles and false positives. In the interest of performance, we decided instead to pick using a single artificial reference (disk) and to forgo such subtraction approaches. To avoid picking these kinds of artifacts, the

user can choose to ignore areas with abnormal local variance or local mean. We find that ignoring high-variance areas often helps avoid edges of problematic objects, e.g. ice crystals or carbon foils, and that avoiding high- and low-mean areas helps avoid picking from areas within them, e.g. the carbon foil itself or within an ice crystal (*Figure 3*). The thresholds used are set to $Mo + 2\ FWHM$ for the variance and $Mo \pm 2\ FWHM$ for the mean, where $Mo$ is the mode (i.e. the most-commonly-occurring value) and $FWHM$ the full width at half-maximum of the distribution of the relevant statistic. For micrographs with additional phase plate phase shifts between 0.1 and 0.9 $\pi$, where much higher contrast is expected, the variance threshold is increased to $Mo + 8\ FWHM$. We have found that in favorable cases many erroneous picks can be avoided. Remaining false-positive picks are removed later during 2D classification.

Because of our emphasis on performance, our algorithm can be run nearly instantaneously on a typical ~4K image, using a single processor. In the Action panel, the user is presented with an 'Auto preview' mode to enable interactive adjustment of the picking parameters (*Figure 3*). In this mode, the micrograph is displayed with optional and adjustable low-pass and high-pass filters, and the results of picking using the currently selected parameters are overlaid on top. Changing one or more of the parameters leads to a fast re-picking of the displayed micrograph, so that the parameters can be optimized in real-time. Once the three main parameters have been adjusted appropriately, the full complement of input micrographs can be picked, usually in a few seconds or minutes.

A possible disadvantage of using a single disk template exists when the particles to be picked are non-uniform in size or shape (e.g. in the case of an elongated particle). In this case, it may be expected that a single template would have difficulty in picking all the different types and views of particles present, and that in this case using a number of different templates would lead to a more

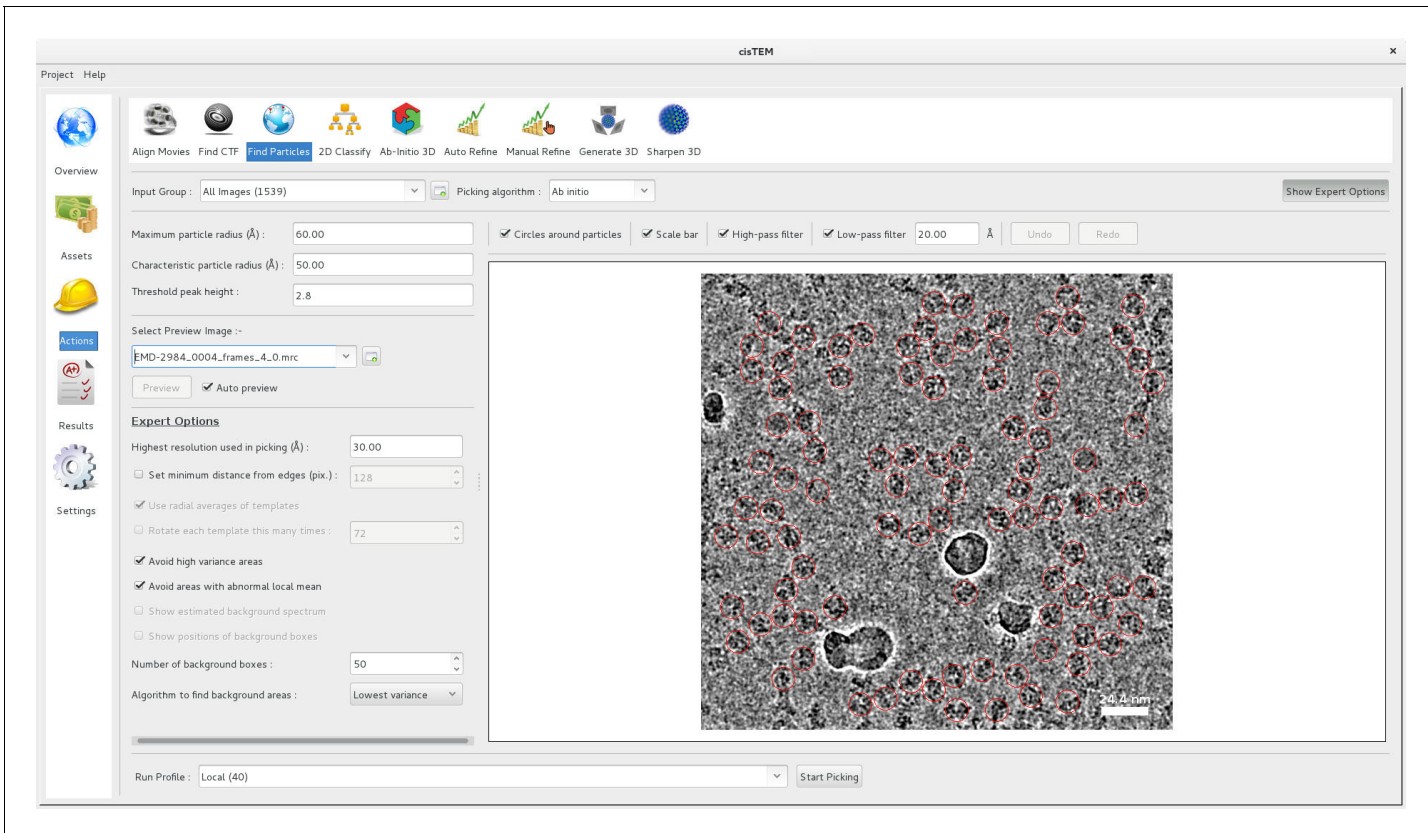

**Figure 3.** Particle picking panel of the *cis*TEM GUI. The panel shows the preview mode, which allows interactive tuning of the picking parameters for optimal picking. The red circles overlaying the image of the sample indicate candidate particles. The picking algorithm avoids areas of high variance, such as the ice contamination visible in the image.

DOI: https://doi.org/10.7554/eLife.35383.004

accurate picking. In practice, we found that with careful optimization of the parameters, elongated particles and particles with size variation (*Figure 3*) were picked adequately.

The underlying implementation of the algorithm supports multiple references as well as reference rotation. These features may be exposed to the graphical user interface in future versions, for example enabling the use of 2D class averages as picking templates (*Scheres, 2015*).

## 2D classification

2D classification is a relatively quick and robust way to assess the quality of a single-particle dataset. *cis*TEM implements a maximum likelihood algorithm (*Scheres et al., 2005*) and generates fully CTF-corrected class averages that typically display clear high-resolution detail, such as secondary structure. Integration of the likelihood function is done by evaluating the function at defined angular steps $d\alpha$ that are calculated according to

$$d\alpha = R/D \tag{1}$$

where $R$ is the resolution limit of the data and $D$ is the diameter of the particle (twice the mask radius that is applied to the iteratively-refined class averages). *cis*TEM runs a user-defined number of iterations $n$ defaulting to 20. To speed up convergence, the resolution limit is adjusted as a function of iteration cycle $l$ ($0 \leq l < n$):

$$R = R_{start} + l\left(R_{finish} - R_{start}\right)/(n-1) \tag{2}$$

where $R_{start}$ and $R_{finish}$ are user-defined resolution limits at the first and last iteration, defaulting to 40 Å and 8 Å, respectively. The user also sets $K$, the number of classes to calculate. Depending on this number and the number of particles $N$ in the dataset, only a percentage $p$ of the particles are included in the calculation. These particles are randomly reselected for each iteration and $p$ is typically small, for example 0.1, in the first 10 iterations ($p_{0-9}$), then increases to 0.3 for iteration 10 to 14 ($p_{10-14}$) and finishes with five iterations including all data ($p_{15-19}$):

$$p_{0-9} = \begin{cases} 300K/N, & 300K/N < 1 \\ 1, & 300K/N \geq 1 \end{cases}$$

$$p_{10-14} = \begin{cases} 0.3, & p_{0-9} < 0.3 \\ p_{0-9}, & p_{0-9} \geq 0.3 \end{cases} \tag{3}$$

$$p_{15-19} = 1.$$

For example, for a dataset containing $N = 100,000$ particles, $p_{0-9} = 0.15$, that is, 15% of the data will be used to obtain $K = 50$ classes. Apart from speeding up the calculation, the stepwise increase of the resolution limit and the random selection of subsets of the data also reduce the chance of overfitting (see also the calculation of ab-initio 3D reconstructions and 3D refinement below) and, therefore, increase the convergence radius of the 2D classification algorithm.

For the calculation of the likelihood function, the particle images $\mathrm{X}_i$ are noise-whitened by dividing their Fourier transforms $\mathcal{F}\{\mathrm{X}_i\}$ by the square root of the radially average noise power spectrum, *NPS*:

$$\mathcal{F}\left\{\tilde{\mathrm{X}}_i\right\}(\mathbf{g}) = \mathcal{F}\{\mathrm{X}_i\}(\mathbf{g})/\sqrt{NPS(g)} \tag{4}$$

where $\mathbf{g}$ is the 2D reciprocal space coordinate and $g = |\mathbf{g}|$ its magnitude. The noise power spectrum is calculated from the boxed particle images using the area outside the circular mask set by the user according to the expected particle size. To increase accuracy, it is further averaged across 2000 randomly selected particles. The background (density outside the mask) is further normalized by adding a constant to each particle that yields a background average of zero.

Finally, at the beginning of each iteration, noise features in the class averages $\mathrm{A}_i$ are suppressed by resetting negative values below a threshold $t_i$ to the threshold:

$$t_i = -0.3\max_j A_{i,j} \tag{5}$$

where $j$ runs over all pixels in average $\mathrm{A}_i$.

### 3D refinement (FrealignX)

The refinement of 3D reconstructions in *cis*TEM uses a version of Frealign (*Lyumkis et al., 2013*) that was specifically designed to work with *cis*TEM. Most of Frealign's control parameters are exposed to the user in the 'Manual Refine' Action panel (*Figure 4*). The 'Auto Refine' and 'Ab-Initio' panels also use Frealign but manage many of the parameters automatically (see below). Frealign's algorithm was described previously (*Grigorieff, 2007*; *Lyumkis et al., 2013*) and this section will mostly cover important differences, including a new objective function used in the refinement, different particle weighting used in reconstructions, optional likelihood-based blurring, as well as new masking options.

### Matched filter

To make Frealign compatible with *cis*TEM's GUI, the code was completely rewritten in C++, and it will be referred to here as Frealign v10, or FrealignX. The new version makes use of a matched filter (*McDonough and Whalen, 1995*) to maximize the signal in cross correlation maps calculated between particle images and reference projections. This requires whitening of the noise present in the images and resolution-dependent scaling of the reference projections to match the signal in the noise-whitened images. Both can be achieved if the spectral signal-to-noise ratio (SSNR) of the data is known. As part of a 3D reconstruction, Frealign calculates the resolution-dependent *PSSNR*, the radially averaged SSNR present in the particle images before they are affected by the CTF

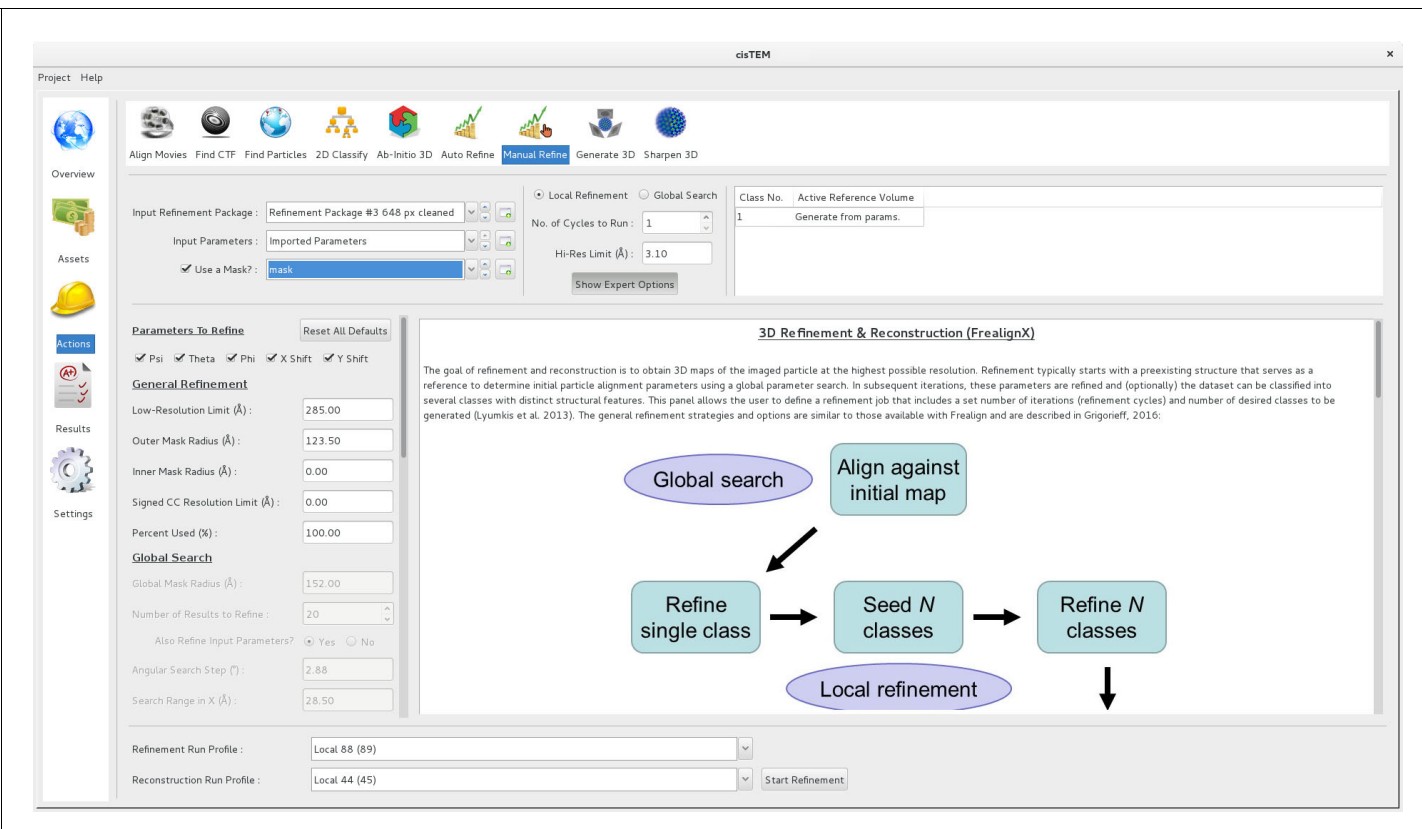

**Figure 4.** Manual refinement panel with Expert Options exposed. Most of the parameters needed to run FrealignX can be accessed on this panel. The panel also allows application of a 3D mask, which can be imported as a Volume Asset.
DOI: https://doi.org/10.7554/eLife.35383.005

(*Sindelar and Grigorieff, 2012*). Using *PSSNR* and the CTF determined for a particle, the SSNR in the particle image can be calculated as

$$SNR(\mathbf{g}) = PSSNR(g) \times CTF^2(\mathbf{g}) \tag{6}$$

(as before, **g** is the 2D reciprocal space coordinate and $g = |\mathbf{g}|$). Here, *SNR* is defined as the ratio of the variance of the signal and the noise. The Fourier transform $\mathcal{F}\{\tilde{\mathrm{X}}_i\}$ of the noise-whitened particle image $\tilde{\mathrm{X}}_i$ can then be calculated as

$$\mathcal{F}\{\tilde{\mathrm{X}}_i\}(\mathbf{g}) = \frac{\mathcal{F}\{\mathrm{X}_i\}(\mathbf{g})}{\sqrt{|\mathcal{F}\{\mathrm{X}_i\}|^2_r(g)}}\sqrt{1 + SNR(\mathbf{g})} \tag{7}$$

where $\mathcal{F}\{\mathrm{X}_i\}$ is the Fourier transform of the original image $\mathrm{X}_i$, $|\cdot|$ is the absolute value, and $|\mathcal{F}\{\mathrm{X}_i\}|^2_r$ is the radially averaged spectrum of the squared 2D Fourier transform amplitudes of image $\mathrm{X}_i$. To implement *Equation (7)*, a particle image is first divided by its amplitude spectrum, which includes power from both signal and noise, and then multiplied by a term that amplifies the image amplitudes according to the signal strength in the image. The reference projection $\mathrm{A}_i$ can be matched by calculating

$$\mathcal{F}\{\tilde{\mathrm{A}}_i\}(\mathbf{g}) = \frac{\mathcal{F}\{\mathrm{A}_i\}(\mathbf{g})}{\sqrt{|\mathcal{F}\{\mathrm{A}_i\}|^2_r(g)}}\sqrt{SNR(\mathbf{g})} \tag{8}$$

*Equation (8)* scales the variance of the signal in the reference to be proportional to the measured signal-to-noise ratio in the noise-whitened images. The main term in the objective function $O(\phi)$ maximized in FrealignX is therefore given by the cross-correlation function

$$CC(\phi) = \frac{Re\left(\mathcal{F}_{R0,R3}\{\tilde{\mathrm{A}}_i(\phi)\}^* \mathcal{F}_{R1,R3}\{\tilde{\mathrm{X}}_i\}\right)}{\left\|\mathcal{F}_{R1,R3}\{\tilde{\mathrm{A}}_i(\phi)\}\right\|\left\|\mathcal{F}_{R1,R3}\{\tilde{\mathrm{X}}_i\}\right\|} \tag{9a}$$

where $\phi$ is a set of parameters describing the particle view, x,y position, magnification and defocus, $Re(\cdot)$ is the real part of a complex number, $\|\cdot\|$ is the Euclidean norm, i.e. the square root of the sum of the squared pixel values, and $\mathcal{F}_{R1,R3}\{\cdot\}^*$ is the conjugate complex value of the Fourier transform $\mathcal{F}_{R1,R3}\{\cdot\}$. The subscripts $R1$ and $R3$ specify the low- and high-resolution limits of the Fourier transforms included in the calculation of *Equation (9a)*, as specified by the user. To reduce noise overfitting, the user has the option to specify also a resolution range in which the absolute value of the cross terms in the numerator of *Equation (9a)* are used (*Grigorieff, 2000*; *Stewart and Grigorieff, 2004*), instead of the signed values (option 'Signed CC Resolution Limit' under 'Expert Options' in the 'Manual Refine' Action panel). In this case

$$CC(\phi) = \frac{Re\left(\mathcal{F}_{R1,R2}\{\tilde{\mathrm{A}}_i(\phi)\}^* \mathcal{F}_{R1,R2}\{\tilde{\mathrm{X}}_i\}\right) + \left|Re\left(\mathcal{F}_{R2,R3}\{\tilde{\mathrm{A}}_i(\phi)\}^* \mathcal{F}_{R2,R3}\{\tilde{\mathrm{X}}_i\}\right)\right|}{\left\|\mathcal{F}_{R1,R3}\{\tilde{\mathrm{A}}_i(\phi)\}\right\|\left\|\mathcal{F}_{R1,R3}\{\tilde{\mathrm{X}}_i\}\right\|} \tag{9b}$$

where $R2$ is specified by the 'Signed CC Resolution Limit.' The objective function also includes a term $R(\phi|\Theta)$ to restrain alignment parameters (*Chen et al., 2009*; *Lyumkis et al., 2013*; *Sigworth, 2004*), which currently only includes the x,y positions:

$$R(\phi|\Theta) = -\frac{\sigma^2}{M}\left(\frac{(x - \bar{x})^2}{2\sigma_x^2} + \frac{(y - \bar{y})^2}{2\sigma_y^2}\right) \tag{10}$$

where $\sigma$ is the standard deviation of the noise in the particle image and $\Theta$ represents a set of model parameters including the average particle positions in a dataset, $\bar{x}$ and $\bar{y}$, and the standard deviations of the x,y positions from the average values, $\sigma_x$ and $\sigma_y$, and $M$ is the number of pixels in the mask applied to the particle before alignment. The complete objective function is therefore

$$O(\phi) = CC(\phi) + R(\phi|\Theta) \tag{11}$$

The maximized values determined in a refinement are converted to particle scores by multiplication with 100.

## CTF refinement

FrealignX can refine the defocus assigned to each particle. Given typical imaging conditions with current instrumentation (300 kV, direct electron detector), this may be useful when particles have a size of about 400 kDa or larger. Depending on the quality of the sample and images, these particles may generate sufficient signal to yield per-particle defocus values that are more accurate than the average defocus values determined for whole micrographs by CTFFIND4 (see above). Refinement is achieved by a simple one-dimensional grid search of a defocus offset applied to both defocus values determined in the 2D CTF fit obtained by CTFFIND4. FrealignX applies this offset to the starting values in a refinement, typically determined by CTFFIND4, and evaluates the objective function, *Equation (11)*, for each offset. The offset yielding the maximum is then used to assign refined defocus values. In a typical refinement, the defocus offset is searched in steps of 50 Å, in a range of ±500 Å. In the case of β-galactosidase (see below), a single round of defocus refinement changed the defocus on average by 60 Å; the RMS change was 80 Å. For this refinement, the resolution for the signed cross terms equaled the overall refinement resolution limit (3.1 Å), i.e. no unsigned cross terms were used. The refinement produced a marginal improvement of 0.05 Å in the Fourier Shell Correlation (FSC) threshold of 0.143, suggesting that the defocus values determined by CTFFIND4 were already close to optimal. In a different dataset of rotavirus double-layer particles, a single round of defocus refinement changed the defocus on average by 160 Å; the RMS change was 220 Å. In this case, the refinement increased the resolution from ~3.0 Å to ~2.8 Å.

## Masking

FrealignX has a 3D masking function to help in the refinement of structures that contain significant disordered regions, such as micelles in detergent-solubilized membrane proteins. To apply a 3D mask, the user supplies a 3D volume that contains positive and negative values. *cis*TEM will binarize this volume by zeroing all voxels with values less than or equal to zero, and setting all other voxels to 1, indicating the region of the volume that is inside the mask. A soft cosine-shaped falloff of specified width (e.g. 10 Å) is then applied to soften the edge of the masked region and avoid sharp edges when the mask is applied to a 3D reconstruction. The region of the reconstruction outside the mask can be set to zero (simple multiplication of the mask volume), or to a low-pass filtered version of the original density, optionally downweighted by multiplication by a scaling factor set by the user. At the edge of the mask, the low-pass filtered density is blended with the unfiltered density inside the mask to produce a smooth transition. *Figure 5* shows the result of masking the reconstruction of an ABC transporter associated with antigen processing (TAP, [*Oldham et al., 2016*]). The mask was designed to contain only density corresponding to protein and the outside density was low-pass filtered at 30 Å resolution and kept with a weight of 100% in the final masked reconstruction. The combination of masking and low-pass filtering in this case keeps a low-pass filtered version of the density outside the mask in the reconstruction, including the detergent micelle. Detergent micelles can be a source of noise in the particle images because the density represents disordered material. However, at low, 20 to 30 Å resolution, micelles generate features in the images that can help in the alignment of the particles. In the case of TAP, this masking prevented noise overfitting in the detergent micelle and helped obtain a reconstruction at 4 Å resolution (*Oldham et al., 2016*).

## 3D reconstruction

In Frealign, a 3D reconstruction $V_k$ of class average $k$ and containing $N$ images is calculated as (*Lyumkis et al., 2013*; *Sindelar and Grigorieff, 2012*)

$$V_k = \mathcal{F}^{-1} \left\{ \frac{\sum_{i=1}^{N} \frac{q_{ik}}{\sigma_i^2} \mathcal{R}\left(\phi_i, \, w_{ik} \cdot CTF_i \cdot \mathcal{F}\left\{\hat{X}_i\right\}\right)}{\sum_{i=1}^{N} \frac{q_{ik}}{\sigma_i^2} \mathcal{R}\left(\phi_i, w_{ik} \cdot CTF_i^2\right) + 1/PSSNR_k} \right\} \tag{12}$$

where $q_{ik}$ is the probability of particle $i$ belonging to class $k$, $\sigma_i$ is the standard deviation of the noise

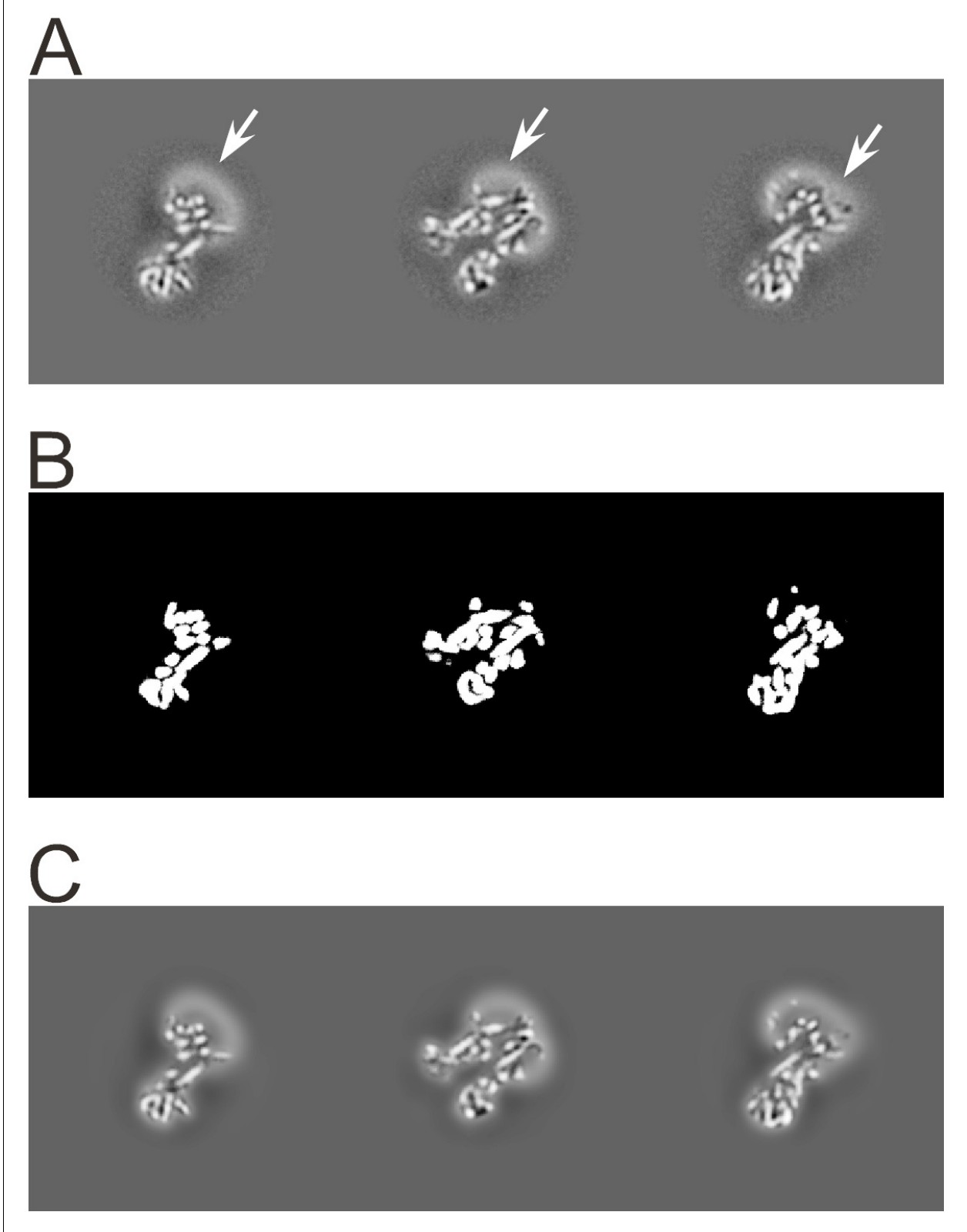

**Figure 5.** 3D masking with low-pass filtering outside the mask. (**A**) Orthogonal sections through the 3D reconstruction of the transporter associated with antigen processing (TAP), an ABC transporter (*Oldham et al., 2016*). Density corresponding to the protein, as well as the detergent micelle (n-Dodecyl b-D-maltoside; highlighted with arrows), is visible. (**B**) Orthogonal sections through a 3D mask corresponding to the sections shown in A). The sharp edges of this mask are smoothed before the mask is applied to the map. (**C**) Orthogonal sections through the masked 3D reconstruction. The

*Figure 5 continued on next page*

*Figure 5 continued*

regions outside the mask are low-pass filtered at 30 Å resolution to remove high-resolution noise from the disordered detergent micelle, but keeping its low-resolution signal to help particle alignment.

DOI: https://doi.org/10.7554/eLife.35383.006

in particle image $i$, $\phi_i$ are its alignment parameters, $w_{ik}$ the score-based weights (*Equation (14)*, see below), $CTF_i$ the CTF of the particle image, $\mathcal{R}(\phi_i, \cdot)$ the reconstruction operator merging data into a 3D volume according to alignment parameters $\phi_i$, $PSSNR$ the radially averaged particle SSNR derived from the FSC between half-maps (*Sindelar and Grigorieff, 2012*), $X_i$ the noise-whitened image $i$, and $\mathcal{F}^{-1}\{\cdot\}$ the inverse Fourier transform. For the calculation of the 3D reconstructions, as well as 3D classification (see below) the particle images are not whitened according to *Equation (7)*. Instead, they are whitened using the radially- and particle-averaged power spectrum of the background around the particles:

$$\mathcal{F}\{\hat{X}_i\}(\mathbf{g}) = \frac{\mathcal{F}\{X_i\}(\mathbf{g})}{\sqrt{\left|\mathcal{F}\{B(X_i)\}\right|^2_r(g)}} \tag{13}$$

where $B(X_i)$ is a masked version of image $X_i$ with the area inside a circular mask centered on the particle replaced with the average values at the edge of the mask, and scaled variance to produce an average pixel variance of 1 in the whitened image $X_i$. Using the procedure in *Equation (13)* has the advantage that whitening does not depend on the knowledge of the SSNR of the data, and reconstructions can therefore be calculated even when the SSNR is not known.

## Score-based weighting

In previous versions of Frealign, resolution-dependent weighting was applied to the particle images during reconstruction (the Frealign parameter was called 'PBC', (*Grigorieff, 2007*)). The weighting function took the form of a B-factor dependent exponential that attenuates the image data at higher resolution. FrealignX still uses B-factor weighting but the weighting function is now derived from the particle scores (see above) as

$$w(score, \mathbf{g}) = e^{-\frac{BSC}{4}\left(score - \bar{score}\right)g^2}. \tag{14}$$

$BSC2$ converts the difference between a particle score and $\bar{score}2$, the score average, into a B-factor. Setting $BSC2$ to zero will turn off score-based particle weighting. Scores typically vary by about 10, and values for $BSC2$ that produce reasonable discrimination between high-scoring and low-scoring particles are between 2 and 10 Å$^2$, resulting in B-factor differences between particles of 20 to 100 Å$^2$.

## 3D classification

FrealignX uses a maximum-likelihood approach for 3D classification (*Lyumkis et al., 2013*). Assuming that all images were noise-whitened according to *Equation (13)*, which scales the variance of each image such that the average standard deviation of the noise in a pixel is 1, the probability density function (PDF) of observing image $X_i$, given alignment parameters $\phi_i$ and reconstruction $V_k$, is calculated as (*Lyumkis et al., 2013*)

$$\Gamma(X_i|\phi_{ik}, V_k) = \left(\frac{1}{2\pi}\right)^{\tilde{M}} \exp\left[-\frac{\left\|\hat{X}_i - CTF_i \cdot \wp(V_k, \phi_{ik})\right\|^2_{\tilde{M}}}{2}\right] \gamma(\phi_{ik}|\Theta_k). \tag{15}$$

As before, $\phi_{ik}$ are the alignment parameters (usually just Euler angles and x,y shifts) determined for image $i$ with respect to class average $k$, $\wp$ is the projection operator producing an aligned 2D projection of reconstruction $V_k$ according to parameters $\phi_{ik}$, $\left\|X_i - CTF_i \cdot \wp(V_k, \phi_{ik})\right\|^2_{\tilde{M}}$ is the sum of the squared pixel value differences between whitened image $X_i$ and the reference projection inside a circular mask defining the area of the particle with user-defined diameter, $\tilde{M}$ is the number of

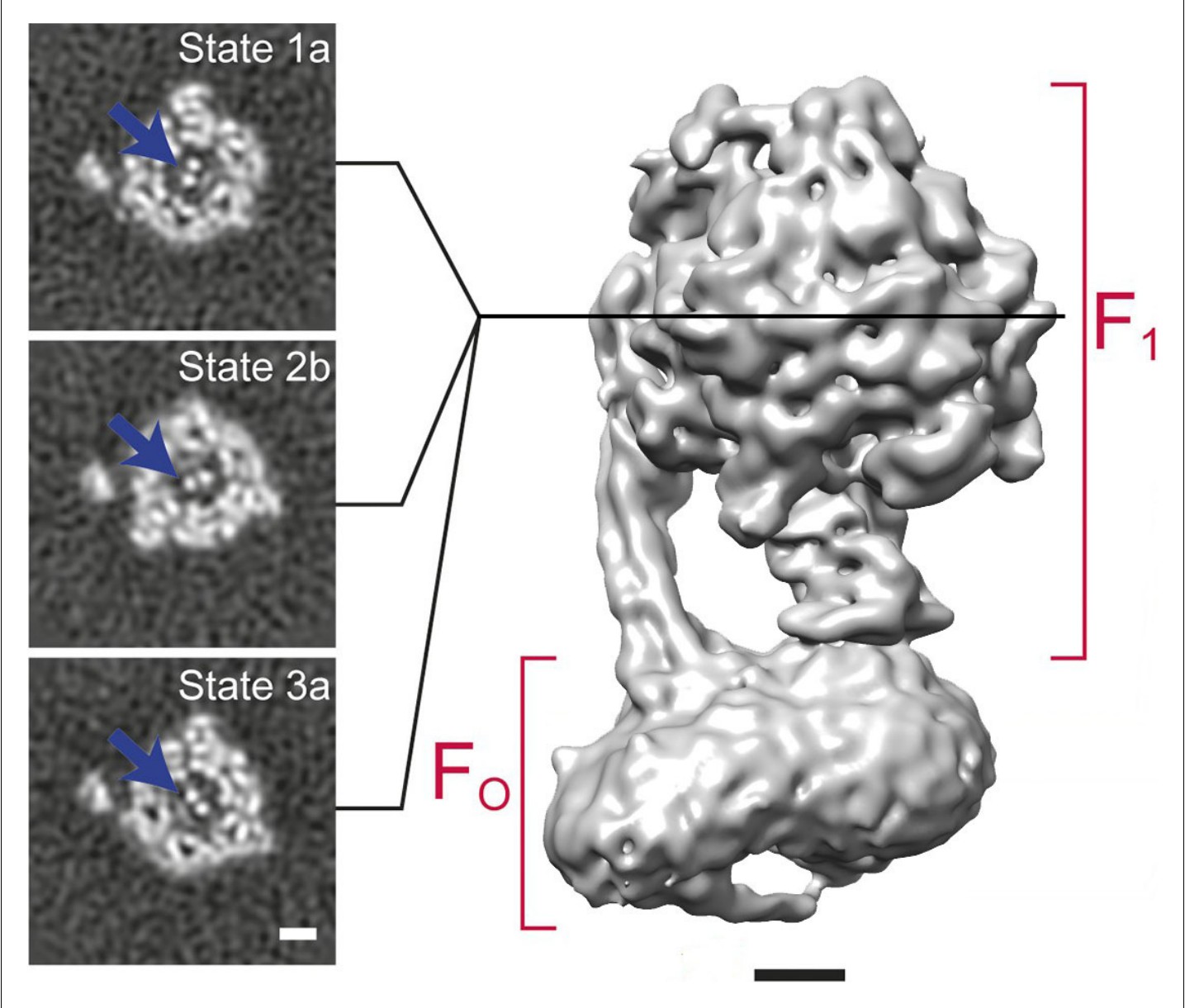

**Figure 6.** 3D classification of a dataset of $F_1F_O$-ATPase, revealing different conformational states (reproduced from *Figure 6A and B* in *Zhou et al., 2015*). Sections through the $F_1$ domain showing the γ subunit (arrows) in three different states related by 120° rotations are shown on the left. A surface rendering of the map corresponding to State 1a is shown on the right. Scale bars, 25 Å.

DOI: https://doi.org/10.7554/eLife.35383.007

pixels inside this mask, and $\gamma(\phi_{ik}|\Theta_k)$ is a hierarchical prior describing the probability of observing alignment parameters $\phi_{ik}$ given model parameters $\Theta_k$ (see *Equation 10*). *Equation (15)* does not include marginalization over alignment parameters. Marginalization could be added to improve classification when particle alignments suffer from significant errors. However, this is currently not implemented in *cis*TEM. Given the joint probability, *Equation (15)*, determined in a refinement, the probability $q_{ik}$ of particle $i$ belonging to class $k$ can be updated as (*Lyumkis et al., 2013*)

$$q_{ik} = \frac{\Gamma(X_i|\Theta_{ik}, V_k)\pi_k}{\sum_{k=1}^{K}\Gamma(X_i|\Theta_{ik}, V_k)\pi_k} \tag{16}$$

where the summation in the denominator is taken over all classes and the average probabilities $\pi_k$

for a particle to belong to class $k$ are given by the average values of $q_{ik}$ determined in a prior iteration, calculated for the entire dataset of $N$ particles:

$$\pi_k = \frac{1}{N}\sum_{i=1}^{N}q_{ik}. \tag{17}$$

An example of 3D classification is shown in *Figure 6* for $F_1F_O$-ATPase, revealing different conformational states of the γ subunit (*Zhou et al., 2015*).

## Focused classification

3D classification can be improved by focusing on conformationally- or compositionally-variable regions of the map. To achieve this, a mask is applied to the particle images and reference projections, the area of which is defined as the projection of a sphere with user-specified center (within the 3D reconstruction) and radius. This 2D mask is therefore defined independently for each particle, as a function of its orientation. When using focused classification, $\tilde{M}$ in *Equation (15)* is adjusted to the number of pixels inside the projected mask and the sum of the squared pixel value differences in *Equation (15)* is limited to the area of the 2D mask. By applying the same mask to image and reference, only variability inside the masked region is used for 3D classification. Other regions of the map are ignored, leading to a 'focusing' on the region of interest. The focused mask also excludes noise contained in the particle images outside the mask and therefore improves classification results that often depend on detecting small differences between particles and references. A typical application of a focused mask is in the classification of ribosome complexes that may exhibit localized conformational and/or compositional variability, for example the variable conformations of an IRES (*Abeyrathne et al., 2016*) or different states of tRNA accommodation (*Loveland et al., 2017*).

## Likelihood-based blurring

In some cases, the convergence radius of refinement can be improved by blurring the reconstruction according to a likelihood function. This procedure is similar to the maximization step in a maximum likelihood approach (*Scheres, 2012*). The likelihood-blurred reconstruction is given by

$$V_k^n = \frac{\sum_{i=1}^{N}\frac{1}{\sigma_i^2}\int_{\phi_{\alpha xy}}\Gamma\left(X_i|\phi_i,V_k^{n-1}\right)\mathcal{R}(\phi_i,\,w_i\cdot CTF_i\cdot X_i)d\phi_{\alpha xy}}{\sum_{i=1}^{N}\frac{q_{ik}}{\sigma_i^2}\mathcal{R}(\phi_i,w_i\cdot CTF_i^2\,)+1/PSSNR_k} \tag{18}$$

where, in the case of FrealignX, $\phi_{\alpha xy}$ only includes the x,y particle positions and in-plane rotation angle $\alpha$, which are a subset of the alignment parameters $\phi_i$, and $V_k^{n-1}$ is the reconstruction from an earlier refinement iteration. As before, $\Gamma\left(X_i|\phi_i,V_k^{n-1}\right)$ is the probability of observing image $i$, given alignment parameters $\phi_i$ and reconstruction $V_k^{n-1}$. Integration over these three parameters can be efficiently implemented and, therefore, does not produce a significant additional computational burden.

## Resolution assessment

The resolution of reconstructions generated by FrealignX is assessed using the FSC criterion (*Harauz and van Heel, 1986*) using the 0.143 threshold (*Rosenthal and Henderson, 2003*). FSC curves in *cis*TEM are calculated using two reconstructions ('half-maps') calculated either from the even-numbered and odd-numbered particles, or by dividing the dataset into 100 equal subsets and using the even- and odd-numbered subsets to calculate the two reconstructions (in the *cis*TEM GUI, the latter is always used). The latter method has the advantage that accidental duplication of particles in a stack is less likely to affect the FSC calculation. All particles are refined against a single reference and, therefore, the calculated FSC values may be biased towards higher values (*Grigorieff, 2000*; *Stewart and Grigorieff, 2004*). This bias extends slightly beyond the resolution limit imposed during refinement, by approximately $2/D_{mask}$, where $D_{mask}$ is the mask radius used to mask the reconstructions (see above). During auto-refinement (see below), the resolution limit imposed during refinement is carefully adjusted to stay well below the estimated resolution of the reconstruction and the resolution estimate is therefore unbiased (*Scheres and Chen, 2012*). However, users have full control over all parameters during manual refinement and will have to make

sure that they do not bias the resolution estimate by choosing a resolution limit that is close to, or higher than, the estimated resolution of the final reconstruction. Calculated FSC curves are smoothed using a Savitzky–Golay cubic polynomial that reduces the noise often affecting FSC curves at the high-resolution end.

The FSC calculated between two density maps is dependent on the amount of solvent included inside the mask applied to the maps. A larger mask that includes more solvent background will yield lower FSC values than a tighter mask. To obtain an accurate resolution estimate in the region of the particle density, one possibility is to apply a tight mask that closely follows the boundary of the particle. This approach bears the risk of generating artifacts because the particle boundary is not always well defined, especially when the particle includes disordered domains that generate weak density in the reconstruction. The approach in Frealign avoids tight masking and instead calculates an FSC curve using generously masked density maps, corrected for the solvent content inside the mask (*Sindelar and Grigorieff, 2012*). The corrected FSC curve is referred to as $Part\_FSC$ and is calculated from the uncorrected $FSC_{uncor}$ as (*Oldham et al., 2016*)

$$Part\_FSC_{half-maps} = \frac{fFSC_{uncor}}{1 + (f-1)FSC_{uncor}}, \tag{19}$$

where $f$ is the ratio of mask volume to estimated particle volume. The particle volume can be estimated from its molecular mass $M_w$ as $\frac{3}{0.81\mathrm{Da}}M_w$ (*Matthews, 1968*). FSC curves obtained with the generous masking and subsequent solvent correction yield resolution estimates that are very close to those obtained with tight masking (*Figure 7C*). *Equation (19)* assumes that both maps have similar SSNR values, as is normally the case for the two reconstructions calculated from two halves of the dataset, indicated by the subscript $half-maps$. If one of the maps does not contain noise from solvent background, for example when calculating the FSC between a reconstruction and a map derived from an atomic model, the solvent-corrected FSC is given as

$$Part\_FSC_{model-map} = \sqrt{\frac{fFSC_{uncor}^2}{1 + (f-1)FSC_{uncor}^2}}. \tag{20}$$

## Speed optimization

FrealignX has been optimized for execution on multiple CPU cores. Apart from using optimized library functions for FFT calculation and vector multiplication (Intel Math Kernel Library), the processing speed is also increased by on-the-fly cropping in real and reciprocal space of particle images and 3D reference maps. Real-space cropping reduces the interpolation accuracy in reciprocal space and is therefore limited to global parameter searches that do not require the highest accuracy in the calculation of search projections. Reciprocal-space cropping is used whenever a resolution limit is specified by the user or in an automated refinement (ab-initio 3D reconstruction and auto-refinement). For the calculation of in-plane rotated references, reciprocal-space padding is used to increase the image size four-fold, allowing fast nearest-neighbor resampling in real space with sufficient accuracy to produce rotated images with high fidelity.

## Ab-initio 3D reconstruction

Ab-initio reconstruction offers a convenient way to proceed from single particle images to a 3D structure when a suitable reference is not available to initialize 3D reconstruction and refinement. Different ab-initio methods have been described (*Hohn et al., 2007*; *Punjani et al., 2017*; *Reboul et al., 2018*) and *cis*TEM's implementation follows a strategy published originally by (*Grigorieff, 2016*). It is based on the premise that iterative refinement of a reconstruction initialized with random angular parameters is likely to converge on the correct structure if overfitting is avoided and the refinement proceeds in small steps to reduce the chance of premature convergence onto an incorrect structure. The procedure is implemented as part of *cis*TEM's GUI and uses FrealignX to perform the refinements and reconstructions.

After initialization with random angles, *cis*TEM performs a user-specified number of global alignment parameter searches, recalculating the reconstruction after each search and applying an automatic masking procedure to it before the next global search. Similar to 2D classification (see above), only a randomly selected subset of the data is used in each iteration and the resolution limit applied

during the search is increased with every iteration. The number of iterations $n$ defaults to 40, the starting and final resolution limits $R_{start}$ and $R_{finish}$ default to 20 Å and 8 Å, respectively, and the starting and final percentage of included particles in the reconstruction, $p_{start}$ and $p_{finish}$ default to $2500K/N$ and $10,000K/N$, respectively (results larger than one are reset to 1), with $K$ the number of 3D classes to be calculated as specified by the user, and $N$ the number of particles in the dataset. If symmetry is applied, $N$ is replaced by $NO_{sym}$ where $O_{sym}$ is the number of asymmetric units present in one particle. The resolution limit is then updated in each iteration $l$ as in *Equation (2)*, and the percentage is updated as

$$p = p_{start} + l\left(p_{finish} - p_{start}\right)/(n-1) \tag{21}$$

again resetting results larger than 1 to 1. *cis*TEM actually performs a global search for a percentage $3p$ of the particle stack, that is, three times as many particles as are included in the reconstructions for each iteration. The particles included in the reconstructions are then chosen to be those with the highest scores as calculated by FrealignX.

The global alignment parameters are performed using the 'general' FrealignX procedure with the following changes. Firstly, the $PSSNR$ is not directly estimated from the FSC calculated at each round. Instead, for the first three iterations, a default $PSSNR$ is calculated based on the molecular weight. From the fourth iteration onwards, the $PSSNR$ is calculated from the FSC, however if the calculated $PSSNR$ is higher than the default $PSSNR$, the default $PSSNR$ is taken instead. This is done in order to avoid some of the overfitting that will occur during refinement. Secondly, during a normal global search the top $h$ (where $h$ defaults to 20) results of the grid search are locally refined, and the best locally refined result is taken. In the ab-initio procedure, however, the result of the global search for a given particle image is taken randomly from all results that have a score which lies in the top 15% of the difference between the worst score and the best score.

During the reconstruction steps, the calculated $\sigma$ for each particle is reset to 1 prior to 3D reconstruction and score weighting is disabled. This is done because the $\sigma$ and score values are not meaningful until an approximately correct solution is obtained.

The reconstructions are automatically masked before each new refinement iteration to suppress noise features that could otherwise be amplified in subsequent iterations. The same masking procedure is also applied during auto-refinement (see below). It starts by calculating the density average $\bar{\rho}$ of the reconstruction and resetting all voxel values below $\bar{\rho}$ to $\bar{\rho}$. This thresholded reconstruction is then low-pass filtered at 50 Å resolution and turned into a binary mask by setting densities equal or below a given threshold $t$ to zero and all others to 1. The threshold is calculated as

$$t = \bar{\rho}_{filtered} + 0.03\left(\bar{\rho}_{\mathrm{max\_500}} - \bar{\rho}_{filtered}\right) \tag{22}$$

where $\bar{\rho}_{filtered}$ is the density average of the low-pass filtered map and $\bar{\rho}_{\mathrm{max\_500}}$ is the average of the 500 highest values in the filtered map. The largest contiguous volume in this binarized map is then identified and used as a mask for the original thresholded reconstruction, such that all voxels outside of this mask will be set to $\bar{\rho}$. Finally, a spherical mask, centered in the reconstruction box, is applied by resetting all densities outside the mask to zero.

The user has the option to repeat the ab-initio procedure multiple times using the result from the previous run as the starting map in each new run, to increase the convergence radius if necessary. In the case of symmetric particles, the default behavior is to perform the first 2/3rds of the iterations without applying symmetry. The non-symmetrized map is then aligned to the expected symmetry axes and the final 1/3rd of the iterations are carried out with the symmetry applied. This default behavior can be changed by the user such that symmetry is always applied, or is never applied.

Alignment of the model to the symmetry axes is achieved using the following process. A brute force grid search over rotations around the x, y and z axes is set up. At each position on the grid the 3D map is rotated using the current x, y and z parameters, and then its projection along Euler angle (0, 0, 0) is calculated. All of the symmetry-related projections are then also calculated, and for each one a cross-correlation map is calculated using the original projection as a reference, and the peak within this map is found. The sum of all peaks from all symmetry-related directions is taken and the x,y,z rotation that most closely aligns the original 3D map along the symmetry axes should provide the highest peak sum. To improve robustness, this process is repeated for two additional angles

(−45,−45, −45 and 15, 70,−15) that were chosen with the aim of including different-looking areas when the map to be aligned is unusual in some way. The x,y,z rotation that results in the largest sum of all peaks, over all three angles, is taken as the final rotation result. Shifts for this rotation are then calculated based on the found 2D x,y shifts between the initial and symmetry-related projections, with the z shift being set to 0 for C symmetries. The symmetry alignment is also included as a command-line program, which can be used to align a volume to the symmetry axis when the ab-initio is carried out in C1 only, or when using a reference obtained by some other means.

## Automatic refinement

Like ab-initio 3D reconstruction, auto-refinement makes use of randomly selected subsets of the data and of an increasing resolution limit as refinement proceeds. However, unlike the ab-initio procedure, the percentage of particles $p_l$ and the resolution limit $R_l$ used in iteration $l$ depend on the resolution of the reconstructions estimated in iteration $l-1$. When the estimated resolution improved in the previous cycle,

$$p_l = \max[p_R, p_{l-1}] \tag{23}$$

with

$$p_R = 8000 K e^{75/R_{l-1}^2}/N \tag{24}$$

where $K$ is the number of 3D classes to be calculated and $N$ the number of particles in the dataset. As before, if the particle has symmetry, $N$ is replaced by $NO_{sym}$ where $O_{sym}$ is the number of asymmetric units present in one particle. If the calculated $p_l$ exceeds 1, it is reset to 1. The resolution limit is estimated as

$$R = FSC_{0.5} - 2/D_{mask} \tag{25}$$

where $FSC_{0.5}$ is the point at which the FSC, unadjusted for the solvent within the mask (see above) crosses the 0.5 threshold and $D_{mask}$ is the user-specified diameter of the spherical mask applied to the 3D reference at the beginning of each iteration, and to the half-maps used to calculate the FSC. The term $2/D_{mask}$ accounts for correlations between the two half-maps due to the masking (see above). When the resolution did not improve in the previous iteration,

$$p_l = 1.5 p_{l-1} \tag{26}$$

(reset to one if resulting in a value larger than 1). At least five refinement iterations are run and refinement stops when $p_l$ reaches 1 (all particles are included) and there was no improvement in the estimated resolution for the last three iterations.

If multiple classes are refined, the resolution limit in *Equation (25)* is set independently for each class, however the highest resolution used for classification is fixed at 8 Å. At least nine iterations are run and refinement stops when $p_l$ reaches 1, the average change in the particle occupancy in the last cycle was 1% or less, and there was no improvement in the estimated resolution for the last three iterations.

In a similar manner to the ab-initio procedure, $\sigma$ values for each particle are set to one and score weighting is turned off. This is done until the refinement resolution is better than 7 Å, at which point it is assumed the model is of a reasonable quality.

## Map sharpening

Most single-particle reconstructions require some degree of sharpening that is usually achieved by applying a negative B-factor to the map. *cis*TEM includes a map sharpening tool that allows the application of an arbitrary B-factor. Additionally, maps can be sharpened by whitening the power spectrum of the reconstruction beyond a user-specified resolution (the default is 8 Å). The whitening amplifies terms at higher resolution similar to a negative B-factor but avoids the over-amplification at the high-resolution end of the spectrum that sometimes occurs with the B-factor method due to its exponential behavior. Whitening is applied after masking of the map, either with a hollow spherical mask of defined inner and outer radius, or with a user-defined mask supplied as a separate 3D volume. The masking removes background noise and makes the whitening of the particle density

more accurate. Both methods can be combined in *cis*TEM, together with a resolution limit imposed on the final reconstruction. The whitened and B-factor-sharpened map can optionally be filtered with a figure-of-merit curve calculated using the FSC determined for the reconstruction (*Rosenthal and Henderson, 2003*; *Sindelar and Grigorieff, 2012*).

## GUI design and workflow

*cis*TEM's GUI required extensive development because it is an integral part of the processing pipeline. GUIs have become more commonplace in cryo-EM software tools to make them more accessible to users (*Conesa Mingo et al., 2018*; *Desfosses et al., 2014*; *Moriya et al., 2017*; *Punjani et al., 2017*; *Scheres, 2012*; *Tang et al., 2007*). Many of the interfaces are designed as so-called wrappers of command-line driven tools, i.e. they take user input and translate it into a command line that launches the tool. Feedback to the user takes place by checking output files, which are also the main repository of processing results, such as movie frame alignments, image defocus measurements and particle alignment parameters. As processing strategies become more complex and the number of users new to cryo-EM grows, the demands on the GUI increase in the quest for obtaining the best possible results. Useful GUI functions include guided user input (so-called wizards) that adjust to specific situations, graphical presentation of relevant results, user interaction with running processes to allow early intervention and make adjustments, tools to manipulate data (e.g. masking), implementation of higher-level procedures that combine more primitive processing steps to achieve specific goals, and a global searchable database that keeps track of all processing steps and result. While some of these functions can be or have been implemented in wrapper GUIs, the lack of control of these GUIs over the data and processes makes a reliable implementation more difficult. For example, keeping track of results from multiple processing steps, some of them perhaps repeated with different parameters or run many times during an iterative refinement, can become challenging if each step produces a separate results file. Communicating with running processes via files can be slow and is sometimes unreliable due to file system caching. Communication via files may complicate the implementation of higher-level procedures, which rely on the parsing of results from the more primitive processing steps.

The *cis*TEM GUI is more than a wrapper as it implements some of the new algorithms in the processing pipeline directly, adjusting the input of running jobs as the refinement proceeds. It enables more complex data processing strategies by tracking all results in a single searchable database. All processing steps are run and controlled by the GUI, which communicates with master and slave processes through TCP/IP. *cis*TEM uses an SQL database, similar to Appion (*Lander et al., 2009*), to store all results (except image files), offers input functions that guide the user or set appropriate defaults, and implements more complex procedures to automate processing where possible. *cis*TEM's design is flexible to allow execution in many different environments, including single workstations, multiple networked workstations and large computer clusters.

User input and the display of results is organized into different panels that make up *cis*TEM's GUI, each panel dedicated to specific processing steps (for examples, see *Figures 1*, *3* and *4*). This design guides users through a standard workflow that most single particle projects follow: movie alignment, CTF determination, particle picking, 2D classification, 3D reconstruction, refinement and classification, and sharpening of the final reconstructions. Three types of panels exist, dealing with Assets, Actions and Results. Assets are mostly data that can be used in processing steps called Actions. They include Movies, Images, Particle Positions and Volumes. One type of Asset, a Refinement Package, defines the data and parameters necessary to carry out refinement of a 3D structure (or a set of structures if 3D classification is done), it contains a particle stack, as well as information about the sample (e.g. particle size and molecular weight) along with parameters for each particle (e.g. orientations and defocus values). Actions comprise the above mentioned workflow steps, with additional options for ab-initio 3D reconstruction, as well as automatic and manual 3D refinement to enable users to obtain the best possible results from their data. The results of most of these Actions are stored in the database and can be viewed in the related Results panels, which display relevant data necessary to evaluate the success of each processing step. The option to sort and select results by a number of different metrics is available in the movie alignment and CTF estimation Results panels. For example images can be sorted/selected based on the CTF fit resolution (*Rohou and Grigorieff, 2015*). Movie alignment, 3D refinement and reconstruction also produce new Image and Volume Assets, respectively.

Importing or generating new Assets is accomplished by wizards that solicit the necessary user input and perform checks were possible to avoid nonsensical input. In the more complex case of creating a new Refinement Package Asset, the wizard allows the specification of input data, for example based on particle picking results or the selection of 2D and 3D classes. Once an Action has been launched, results are displayed as soon as they become available, together with an overall progress bar, giving users an estimate of how long a processing step will take and of whether the results are as expected. If desired, an Action can be aborted and restarted with a different set of parameters, or the Action can be run again after regular termination to test different parameters. In the latter case, all prior results remain accessible and users can specify those to be used for the next step in the workflow. This provides users with the flexibility to pick and choose the best results in cases where different parts of a dataset require different settings to yield optimal results.

## Parallelization

*cis*TEM uses a home-grown scheme to accelerate processing in parallel environments. Image processing of single-particle data is an embarrassingly parallel problem, i.e. the parallelization of most tasks can be achieved simply by dividing the data to be processed into smaller chunks that are each processed by a separate program thread, without the need for inter-process communication. Only certain steps require merging of data, such as the calculation of a 3D reconstruction from the entire dataset. *cis*TEM parallelizes processing steps by running multiple instances of the same program, each dealing with a subset of the data, then directly communicating with the launched processes over TCP/IP sockets. This enables the *cis*TEM GUI to distribute jobs and receive results in real time. Communication is directed through a 'manager' process, which enables jobs to be run on a cluster, while the GUI itself can run on a local workstation.

How each copy of the program is run is specified by the user by setting up a 'Run Profile'. This profile is a user defined command, or script that will be run to launch the job, and is designed to be flexible to enable the user to set up parallelization in many different environments. For example, users can design profiles to run on multiple machines via SSH, or to submit to a cluster (e.g. using qsub) etc., or even merge the two in a single profile. One disadvantage of this system is that it may be difficult to create profiles for clusters that require many jobs to be submitted using one command.

Another advantage of using a home-grown scheme over existing schemes (e.g. MPI) occurs when jobs are run on a multi-node computing cluster. In this case, jobs will complete even if the full number of requested processors is not available. For example, if a user requests 300 CPUs for a processing step but only 100 are available, *cis*TEM launches 300 jobs of which 200 will remain in the job scheduler's queue. Data processing starts immediately with the 100 jobs that are allowed to run and will complete even if the remaining jobs never run. In such a scenario, an MPI-based job could only run when 300 CPUs become available, potentially delaying execution. In the few cases were a step requires merging of an entire dataset, for example in a 3D reconstruction, parallelization is achieved by calculating multiple intermediate 3D reconstructions for subsets of the data, dumping the intermediate reconstructions to disk and merging them after all reconstruction jobs have finished. It can therefore help to designate a fast disk as a scratch disk to allow rapid dumping and reading of the relatively large files (200 MB – 5 GB).

## Benchmarking with β-galactosidase

A high-resolution dataset of β-galactosidase (*Bartesaghi et al., 2015*) was used to benchmark Relion 2 (*Kimanius et al., 2016*) and is also used here to illustrate the workflow of *cis*TEM and assess the time for the completion of different processing steps. The entire dataset was downloaded from the EMPIAR database (*Iudin et al., 2016*) and consists of 1539 movies containing 38 frames, recorded at super-resolution on a K2 Summit camera (Gatan, Inc., Pleasanton, CA) and stored locally as tif files using LZW compression (the conversion to tiff and compression was performed by mrc2tif (*Mastronarde and Held, 2017*)). The pixel size of the super-resolution frames was 0.3185 Å, and images were binned to a pixel size of 0.75 Å after movie processing. For 2D classification and ab-initio 3D reconstruction, particles were boxed using 384 × 384 pixel boxes. For auto- and manual refinement, the particles were re-boxed into 648 × 648 pixel boxes (boxing is part of the creation of Refinement Packages, see above). For all processing steps, a Dell Precision T7910 workstation

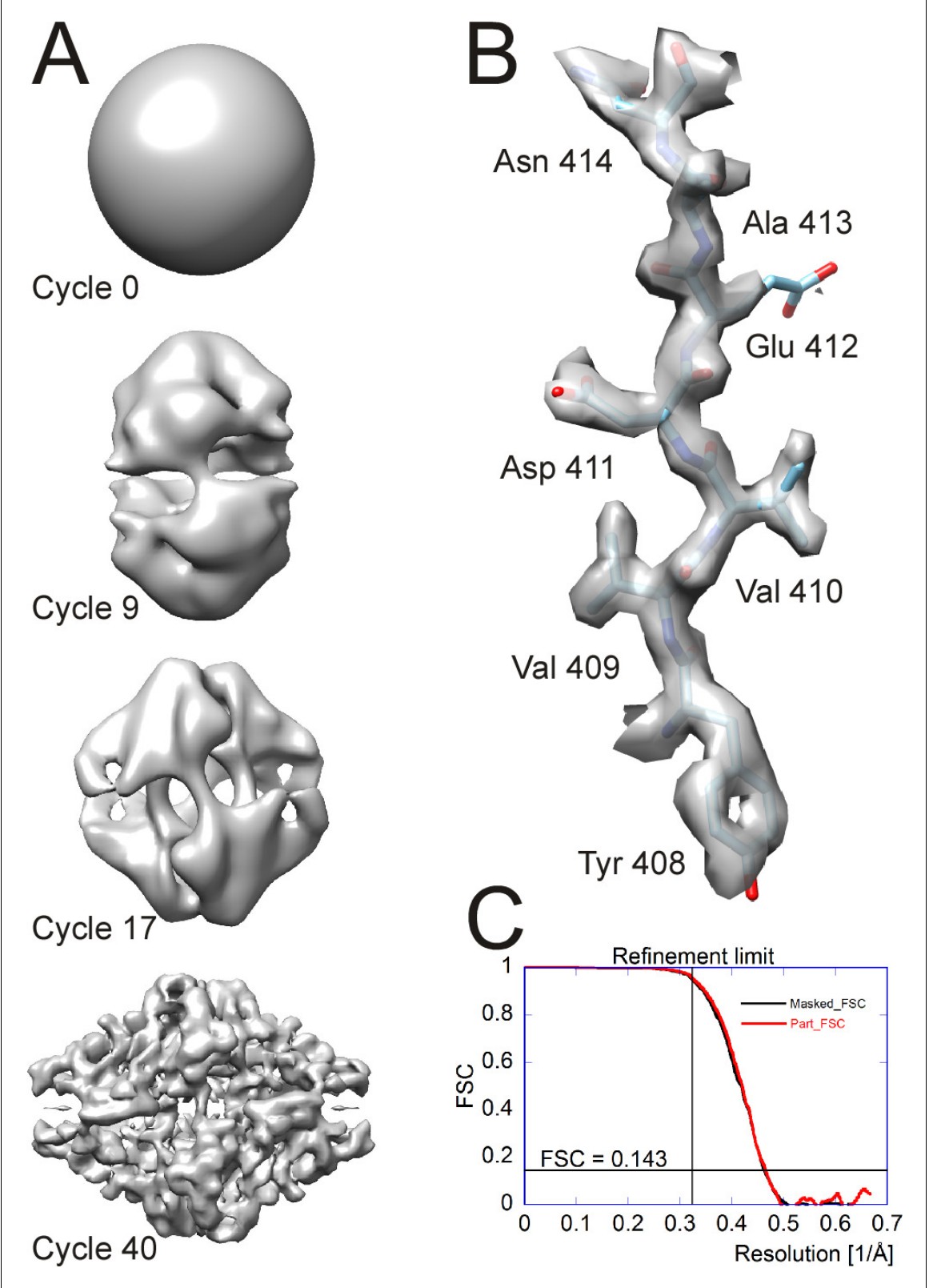

**Figure 7.** Processing results of the β-galactosidase dataset (*Bartesaghi et al., 2015*) used to benchmark *cis*TEM. (**A**) Different stages of the ab-initio reconstruction procedure, starting from a reconstruction from randomly assigned Euler angles. The process takes less than an hour to complete on a high-end CPU-based workstation. (**B**) High-resolution detail of the refined β-galactosidase reconstruction with an average resolution of 2.2 Å, showing sidechain details for most amino acids. (**C**) FSC plots for the refined β-galactosidase reconstruction. The black curve was calculated using a tight mask

*Figure 7 continued on next page*

*Figure 7 continued*

applied to the half maps (Masked_FSC). A correction for potential masking artifacts (*Chen et al., 2013*) did not lead to adjustments of this curve. The red curved was calculated with a more generous spherical mask and adjusted for the solvent background within that mask (Part_FSC, *Equation (19)*). The resolution limit of 3.1 Å, which was not exceeded during refinement, as well as the FSC = 0.143 threshold are indicated by lines.

DOI: https://doi.org/10.7554/eLife.35383.008

The following source data is available for figure 7:

**Source data 1.** Source data for the curves shown in *Figure 7C*.

DOI: https://doi.org/10.7554/eLife.35383.009

containing two E5-2699 v4 Xeon processors with a total of 44 cores was used. Processing parameters were left on default settings except for CTF determination, which was performed at 3.5 Å resolution using the movie averages instead of the frames, and particle picking, which used optimized parameters based on previewing a few selected images (*Figure 3*). The resolution limit during refinement (auto, manual and CTF) never exceeded 3.1 Å. The data were stored on a local SSD Raid 0 disk for fast access. *Table 1* lists the timings of the different processing steps using all 44 CPU cores. Results obtained at different points in the workflow are shown in *Figure 7*.

## Discussion

The implementation of a complete image processing workflow in *cis*TEM offers users a uniform experience and guarantees smooth transitions between processing steps. It also helps developers maintain the software as all the tools and algorithms are developed in-house.

The main focus of *cis*TEM is on the processing of single-particle cryo-EM data and high-resolution 3D reconstruction. Future releases of *cis*TEM may include on-the-fly processing of data as it is collected, particle-based movie alignment, support for helical particles, improved 3D masking tools, more reliable resolution and quality indicators, as well as miscellaneous tools such as the determination of the detective quantum efficiency of electron detectors.

Since *cis*TEM does not rely on third-party libraries, such as Python, MPI or CUDA, that usually have to be installed and compiled separately on the target system, ready-to-run binaries can be made available for download that are optimized for different architectures. Using the wxWidgets library also means that *cis*TEM can be compiled for different operating systems, including Linux, Windows and OSX. Using a configure script, different options for the fast Fourier transforms (FFTs) can be specified, including the FFTW (http://www.fftw.org) and Intel MKL (http://software.intel.com/en-us/mkl) libraries. The downloadable binaries are statically linked against the MKL library as this exhibits superior speeds compared to the FFTW library on Intel-based CPUs.

While the lack of support for GPUs simplifies the installation and execution of *cis*TEM, it can also be a limitation on workstations that are optimized for GPU-accelerated code. These workstations often do not have many CPU cores and, therefore, *cis*TEM will run significantly more slowly than

**Table 1.** Benchmarking of *cis*TEM using a high-resolution dataset of β-galactosidase (*Bartesaghi et al., 2015*).

| Processing step | Details | Time (hours) |
| --- | --- | --- |
| Movie processing | 1539 movies, 38 frames, super-resolution | 1.1 |
| CTF determination | Using aligned movie average as input | 0.1 |
| Particle picking | 131,298 particles | 0.1 |
| 2D classification | 50 classes, 28 selected with 119,523 particles | 0.8 |
| Ab initio 3D reconstruction | 40 iterations | 0.8 |
| Auto refinement | 8 iterations, final resolution 2.2 Å | 1.4 |
| Manual refinement | 1 iteration (incl. defocus), final resolution 2.2 Å | 0.4 |
| Total | | 4.7 |

DOI: https://doi.org/10.7554/eLife.35383.010

code that can take advantage of the GPU hardware. Users who would like to run both CPU and GPU-optimized software may therefore have to invest in both types of hardware.

## Materials and methods

### Development of *cis*TEM
The entire *cis*TEM image processing package was written in C++ using the wxWidgets toolkit (http://wxwidgets.org) to implement the GUI, as well as the libtiff library (http://www.libtiff.org) to support the tiff image format, the SQLite library (https://sqlite.org) to implement the SQL database, and Intel's MKL library (http://software.intel.com/en-us/mkl) for the calculation of Fourier transforms and vector products. Optionally, *cis*TEM can also be linked against the FFTW library (http://www.fftw.org) to replace the MKL library. The code was written and edited using Eclipse (http://www.eclipse.org), and GitHub (http://github.com) was used for version control. This code is available in *Source code 1*.

### Benchmark dataset
The performance of *cis*TEM was benchmarked using a cryo-EM dataset of β-galactosidase (*Bartesaghi et al., 2015*), entry EMPIAR-10061 in the EMPIAR database (*Iudin et al., 2016*).

### Image and data formats
*cis*TEM stores all image data using the MRC format (*Crowther et al., 1996*). Additionally, particle parameters can be imported from, and exported to Frealign (*Grigorieff, 2016*) and Relion (*Scheres, 2012*).

## Acknowledgements
The authors are grateful for feedback from early testers of *cis*TEM, including Ruben Diaz-Avalos, Sarah Loerch, Priyanka Abeyrathne, Peter Rickgauer, Ben Himes, Andrei Korostelev, Anna Loveland, Gabriel Demo, Jue Chen, Dmitry Lyumkis, Hiro Furukawa, Wei Lu and Juan Du.

## Additional information

### Competing interests
Nikolaus Grigorieff: Reviewing editor, *eLife*. The other authors declare that no competing interests exist.

### Funding

| Funder | Author |
| --- | --- |
| Howard Hughes Medical Institute | Timothy Grant<br>Alexis Rohou<br>Nikolaus Grigorieff |

The funders had no role in study design, data collection and interpretation, or the decision to submit the work for publication.

### Author contributions
Timothy Grant, Alexis Rohou, Conceptualization, Software, Formal analysis, Supervision, Validation, Investigation, Visualization, Methodology, Writing—original draft, Writing—review and editing; Nikolaus Grigorieff, Conceptualization, Software, Formal analysis, Supervision, Funding acquisition, Validation, Investigation, Visualization, Methodology, Writing—original draft, Project administration, Writing—review and editing

Author ORCIDs
Alexis Rohou https://orcid.org/0000-0002-3343-9621
Nikolaus Grigorieff https://orcid.org/0000-0002-1506-909X

Decision letter and Author response
Decision letter https://doi.org/10.7554/eLife.35383.016
Author response https://doi.org/10.7554/eLife.35383.017

## Additional files

### Supplementary files

• Source code 1. Source code for *cis*TEM 1.0 beta.
DOI: https://doi.org/10.7554/eLife.35383.011

• Transparent reporting form
DOI: https://doi.org/10.7554/eLife.35383.012

### Major datasets

The following previously published dataset was used:

| Author(s) | Year | Dataset title | Dataset URL | Database, license, and accessibility information |
|---|---|---|---|---|
| Bartesaghi A, Merk A, Banerjee S, Matthies D, Wu X, Milne JL, Subramaniam S | 2015 | 2.2 A resolution cryo-EM structure of beta-galactosidase in complex with a cell-permeant inhibitor | https://www.ebi.ac.uk/pdbe/emdb/empiar/entry/10061 | Publicly available at the Electron Microscopy Public Image Archive (accession no. 10061) |

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
