## [Decision Letter]

Thank you for submitting your article "*cis*TEM: User-friendly software for single-particle image processing" for consideration by *eLife*. Your article has been reviewed by three peer reviewers, and the evaluation has been overseen by a Reviewing Editor (Ed Egelman) and John Kuriyan as the Senior Editor. The following individuals involved in review of your submission have agreed to reveal their identity: Sjors HW Scheres (Reviewer #2); Henning Stahlberg (Reviewer #3); Bridget Carragher (Reviewer #4).

The reviewers have discussed the reviews with one another and the Reviewing Editor has drafted this decision to help you prepare a revised submission.

Summary

This paper describes a new image processing software solution for cryo-EM single-particle analysis called *cis*TEM. *cis*TEM combines many existing and very popular tools from the Grigorieff lab (CTFFIND, UNBLUR, FREALIGN) with several new functionalities, such as auto-picking, 2D classification and ab initio model calculation. The existing tools were all very popular solutions in the field, and their inclusion into *cis*TEM has motivated a complete re-write in C++. By careful code optimisation, the new programs now run at speeds that are comparable with (or even surpass) implementations by others on GPUs (e.g. RELION or cryoSPARC). In line with their excellent track-record, the authors release the code as open-source, which is laudable in a time where many new software implementations have moved towards licensing solutions for non-academic users. Modifications to existing methods and new methods are described in sufficient detail to be useful for potential users, and in many cases even for those programmers eager to implement their own versions. The GUI described appears to be well thought-through and thereby a good help for less experienced users. This is particularly relevant for access to the 3D classification and refinement approaches, which were less accessible in previous versions of FREALIGN. The results presented for β-gal are impressive: a 2.2A reconstruction is state-of-the-art, and the execution time below 5 hours is extremely fast! Therefore, this tool will undoubtedly be extremely useful for the field. The performance of *cis*TEM is amazing, in terms of resolution and speed. The authors are to be congratulated for making such a wonderful package available as open-source.

Essential revisions:

1) Eq 15 does not have a CTF_i component. Is that part of the projection operator (with the funny P)? If so, that would be good to spell out. Otherwise, CTF_i should be added to the equation.

2) It was unclear whether the 3D classification approach only marginalises over classes, or whether the marginalisation over the 3 in-plane orientations can also be performed. Perhaps an extra sentence would help.

3) Is *cis*TEM capable of on-the-fly processing? If so, it might be worth spelling that out, as many EM centres in the world are moving towards automated processing of images while they are being acquired.

4) In subsection “Parallelization”, the authors describe a flexible home-made solution to queueing many little sub-jobs, which could provide additional flexibility on busy clusters. However, it might very well be that many clusters require a given number of free nodes to be specified in their own queueing system in order to use the cluster. Perhaps an additional sentence about this could be added?

5) *cis*TEM is available under the GNU General Public License. This is not stated in this manuscript but should be.

6) Introduction: Use "direct electron detectors", not "direct detectors".

7) Subsection “Particle picking”: Please define "mode" in this context.

8) Subsection “Particle picking”: It is obvious that the "need" is there now for external users. This statement could be rephrased.

9) Subsection “3D refinement (FrealignX)”: The size threshold of 400kDa, up to which individual particle defocus refinement might be helpful, will depend on various other parameters, such as kV, defocus, ice thickness, electron detector, etc. This statement could be extended to state that this threshold applies to current typical hardware and sample preparation methods.

10) Subsection “3D refinement (FrealignX)”: For the description of applications of particle-wise CTF refinements, it would be helpful to specify, if this was done using the absolute values of the CCterms in Equation (9b), and if so, which Resolution R2 was used as transition from signed to unsigned CC values.

Also, how much does the particle-wise CTF refinement improve the final resolution, if the signed versus the unsigned CC calculation is used?

11) Subsection “3D refinement (FrealignX)”: FSC calculation is done for particles aligned against a single reference, which is resolution limited to a value well below the current resolution estimate minus 2/Dmask. However, here the authors also state that the users can set the resolution limit of the reference to arbitrary values, at their own risk. This sounds like a dangerous option for unexperienced users. Does *cis*TEM in such cases print out a big fat warning somewhere?

12) Subsection “Ab-initio 3D reconstruction”: "3 direction are chosen". ("s" is missing in "directions").

13) Subsection “Ab-initio 3D reconstruction”: What are the 3 directions that are chosen here? Are those random, linearly independent directions? Or are these related to symmetry axes? Please specify or explain better.

14) Discussion section: GPU hardware doesn't have to be noisier that CPU hardware. Water-cooled housings and water-cooled GPU cards are available, even though at higher costs. Several high-end GPU systems are absolutely silent, even under load. This statement is only valid under very specific settings. I would remove or rephrase it.

15) It would be helpful to know how projects are tracked in a multi user facility. In some facilities there are 100's of different projects, many of which must be well separated for confidentiality reasons. How does *cis*TEM handle this?

16) Can new programs easily be incorporated into *cis*TEM? For example, on one of the few features lacking right now is per particle unblurring and I suspect that many people will want to go with motioncorr2. While the authors discuss the intimate link between the GUI and the code as a positive, it would also be good if there were some options to "wrap" other programs as needed. If not, I suspect users will continue to leap in and out of a multitude of packages.

17) Does the package provide a way for sorting particles or images based on confidence values or outputs. E.g. reject all images where the CTF fit is poor etc.

18) While I accept that the DoG algorithm can be modified to select β-gal I would like to know if it really can be optimized on something much more stick like. If not, then I suspect users will have to exit the program to use a template picker until one becomes available. Again, pointing to the value of wrapping one of the many very good available template pickers.

19) It has been discussed by Gabe Lander (and other groups, too) that there is an advantage in packing particles very closely together across holes (e.g. see the Aldolase images in https://www.biorxiv.org/content/early/2017/05/25/141994). How will this affect programs that need to select background regions for determining optimal particle picking parameters or for noise whitening etc.

20) Per particles defocus is also likely to be very useful for images that are tilted relative to the electron beam or that are in two layers (see for example: https://www.biorxiv.org/content/early/2017/12/11/230276). Could the per particle be done in patches? How do the *cis*TEM results compare to those of gCTF for small particles? Is it possible that the lack of improvement for β-gal compared to the virus is because the method just failed due to the lower signal? How would a user know this? One way might be to plot the defocus values as z-heights which if particles really do all adhere to an air water interface will be in a plane. This might also help smooth out outliers.

21) Should cryoSparc be part of the initial set of software referenced in the Introduction.

22) What FSC is reported, 0.143?

23) Masking instructions are a bit confusing (Subsection “3D refinement (FrealignX)”).

24) Subsection “3D refinement (FrealignX)”: What happened without the mask?

25) BSC scoring (Subsection “3D refinement (FrealignX)”) description is a bit vague.

26) How does the masking affect the FSC exactly – does it come out the same?

27) Subsection “Map sharpening” calculate should be calculated (and this was the only typo I noticed!)

28) Database is interesting, but it is not clear if is it useful to the end user or just the programmers. It might be polite to reference Appion which has been using a very effective database as a tool for delivering results to users and wrapping disparate software packages for over a decade.

29) Can *cis*TEM be run remotely? Does your computer have to stay awake during a long action?

---

## [Author Response]

Essential revisions:1) Eq 15 does not have a CTF_i component. Is that part of the projection operator (with the funny P)? If so, that would be good to spell out. Otherwise, CTF_i should be added to the equation.

Done.

2) It was unclear whether the 3D classification approach only marginalises over classes, or whether the marginalisation over the 3 in-plane orientations can also be performed. Perhaps an extra sentence would help.

Done (see added text, Subsection “3D refinement (FrealignX)).

3) Is *cis*TEM capable of on-the-fly processing? If so, it might be worth spelling that out, as many EM centres in the world are moving towards automated processing of images while they are being acquired.

This is currently not implemented but on our to-do list (see added text, Discussion section).

4) In subsection “Parallelization”, the authors describe a flexible home-made solution to queueing many little sub-jobs, which could provide additional flexibility on busy clusters. However, it might very well be that many clusters require a given number of free nodes to be specified in their own queueing system in order to use the cluster. Perhaps an additional sentence about this could be added?

The current run profiles are designed to be flexible in that the user can provide any command or script they choose to run the command. In some cases, the user may prefer to have one command to run many processes, e.g. using mpirun. This is not easily achievable in the current setup, although this ability will likely be added in a future version. We have added a paragraph (subsection “Parallelization”) that discusses the job submission and this limitation in more detail.

5) *cis*TEM is available under the GNU General Public License. This is not stated in this manuscript but should be.

*cis*TEM is distributed under the Janelia Research Campus Software License (http://license.janelia.org/license/janelia_license_1_2.html) (see added text, Introduction).

6) Introduction: Use "direct electron detectors", not "direct detectors".

Done.

7) Subsection “Particle picking”: Please define "mode" in this context.

The mode of a distribution is the location of the peak of that distribution. Text has been modified (subsection “Particle picking”).

8) Subsection “Particle picking”: It is obvious that the "need" is there now for external users. This statement could be rephrased.

Done (see text, subsection “Particle picking”).

9) Subsection “3D refinement (FrealignX)”: The size threshold of 400kDa, up to which individual particle defocus refinement might be helpful, will depend on various other parameters, such as kV, defocus, ice thickness, electron detector, etc. This statement could be extended to state that this threshold applies to current typical hardware and sample preparation methods.

Done (see text, subsection “3D refinement (FrealignX)”).

10) Subsection “3D refinement (FrealignX)”: For the description of applications of particle-wise CTF refinements, it would be helpful to specify, if this was done using the absolute values of the CCterms in Equation (9b), and if so, which Resolution R2 was used as transition from signed to unsigned CC values.Also, how much does the particle-wise CTF refinement improve the final resolution, if the signed versus the unsigned CC calculation is used?

Absolute CC terms were not used (see added text, subsection “3D refinement (FrealignX)”).

11) Subsection “3D refinement (FrealignX)”: FSC calculation is done for particles aligned against a single reference, which is resolution limited to a value well below the current resolution estimate minus 2/Dmask. However, here the authors also state that the users can set the resolution limit of the reference to arbitrary values, at their own risk. This sounds like a dangerous option for unexperienced users. Does *cis*TEM in such cases print out a big fat warning somewhere?

We will consider adding a general warning to the Manual Refinement panel in the next release. The risk of biased FSC is only one of several that an inexperienced user is exposed to on this panel.

12) Subsection “Ab-initio 3D reconstruction”: "3 direction are chosen". ("s" is missing in "directions").

Done.

13) Subsection “Ab-initio 3D reconstruction”: What are the 3 directions that are chosen here? Are those random, linearly independent directions? Or are these related to symmetry axes? Please specify or explain better.

The three sets of Euler angles used in the format (psi, theta, phi) are (0, 0, 0), (-45, -45, -45) and (15, 70, -15), which were chosen to be sufficiently different to improve robustness of the algorithm. In theory the method should work with any angle, and only one should be needed. In our method we use three and average over the results, decreasing the chance of failure due to a pathologically oriented map, however we have not tested this to determine whether it is needed. We have changed the text to add more detail (subsection “Ab-initio 3D reconstruction”).

14) Discussion section: GPU hardware doesn't have to be noisier that CPU hardware. Water-cooled housings and water-cooled GPU cards are available, even though at higher costs. Several high-end GPU systems are absolutely silent, even under load. This statement is only valid under very specific settings. I would remove or rephrase it.

Removed.

15) It would be helpful to know how projects are tracked in a multi user facility. In some facilities there are 100's of different projects, many of which must be well separated for confidentiality reasons. How does *cis*TEM handle this?

*cis*TEM keeps all results of a project in its project directory. In a multi-user facility where confidentiality is important, users can limit read access to this directory to prevent unauthorized access.

16) Can new programs easily be incorporated into *cis*TEM? For example, on one of the few features lacking right now is per particle unblurring and I suspect that many people will want to go with motioncorr2. While the authors discuss the intimate link between the GUI and the code as a positive, it would also be good if there were some options to "wrap" other programs as needed. If not, I suspect users will continue to leap in and out of a multitude of packages.

Unfortunately, *cis*TEM’s current design does not allow easy incorporation of other software. cisTEM enables data import and export at all stages, and it should be easy for users to perform motion correction in MotionCorr2, and then import the aligned sums for further processing in *cis*TEM.

17) Does the package provide a way for sorting particles or images based on confidence values or outputs. E.g. reject all images where the CTF fit is poor etc.

The option to sort results is available in some panels, most notably in the CTF estimation Results panel, where the images can be sorted / selected based on the CTF fit. Due to the fact the results are stored in a database, sorting / selecting is relatively easy to implement, and future versions will have more sorting facilities. We have added a brief description of this in subsection “GUI design and workflow”.

18) While I accept that the DoG algorithm can be modified to select β-gal I would like to know if it really can be optimized on something much more stick like. If not, then I suspect users will have to exit the program to use a template picker until one becomes available. Again, pointing to the value of wrapping one of the many very good available template pickers.

A simple disk as a template will perform increasingly worse as the shape of the particle becomes more stick-like. As indicated in the manuscript, a template-based picker is in the planning. In the meantime, particle coordinates found by a different picker can be imported into *cis*TEM. We have added a reference to the DoG algorithm (subsection “Particle picking”).

19) It has been discussed by Gabe Lander (and other groups, too) that there is an advantage in packing particles very closely together across holes (e.g. see the Aldolase images in https://www.biorxiv.org/content/early/2017/05/25/141994). How will this affect programs that need to select background regions for determining optimal particle picking parameters or for noise whitening etc.

Our particle picking algorithm will indeed be susceptible to misestimating the background noise spectrum under such circumstances. Picking will still work, but a higher false-positive rate might be expected. This is now spelt out in the text (subsection “Particle picking”).

20) Per particles defocus is also likely to be very useful for images that are tilted relative to the electron beam or that are in two layers (see for example: https://www.biorxiv.org/content/early/2017/12/11/230276). Could the per particle be done in patches? How do the *cis*TEM results compare to those of gCTF for small particles? Is it possible that the lack of improvement for β-gal compared to the virus is because the method just failed due to the lower signal? How would a user know this? One way might be to plot the defocus values as z-heights which if particles really do all adhere to an air water interface will be in a plane. This might also help smooth out outliers.

The lack of improvement in the case of β-galactosidase could certainly be due to inaccuracies in the defocus refinement. CTF refinement in patches, as it is already implemented in gCTF, is an excellent idea. A simplified patch-based refinement was implemented in Frealign 9 (a “patch” consisted of a whole micrograph). We will add such a feature in a future release of *cis*TEM.

21) Should cryoSparc be part of the initial set of software referenced in the Introduction.

Done.

22) What FSC is reported, 0.143?

Yes (see added text, subsection “3D refinement (FrealignX)”).

23) Masking instructions are a bit confusing (Subsection “3D refinement (FrealignX)”).

We rewrote the description of the masking procedure (see changed text, Subsection “3D refinement (FrealignX)”).

24) Subsection “3D refinement (FrealignX)”: What happened without the mask?

Without masking, the detergent micelle showed typical signs of noise overfitting (see added text, Subsection “3D refinement (FrealignX)”).

25) BSC scoring (Subsection “3D refinement (FrealignX)”) description is a bit vague.

We added an explanation of how much the effective B-factors used for weighting vary in a typical case (see added text, Subsection “3D refinement (FrealignX)”)).

26) How does the masking affect the FSC exactly – does it come out the same?Yes, the more generous masking and subsequent solvent correction yields FSC curves that are very close to curves obtained with tightly masked half volumes (see added text, Subsection “3D refinement (FrealignX)”, caption and added curve, Figure 7).27) Subsection “Map sharpening” calculate should be calculated (and this was the only typo I noticed!).

Done.

28) Database is interesting, but it is not clear if is it useful to the end user or just the programmers. It might be polite to reference Appion which has been using a very effective database as a tool for delivering results to users and wrapping disparate software packages for over a decade.

Some of the sorting functions (see above) depend on the database. Also, results keep a record of which input data were used to generate them. These functions are useful to both programmers and users. We have added a reference to Appion (Subsection “GUI design and workflow”).

29) Can *cis*TEM be run remotely? Does your computer have to stay awake during a long action?

Since *cis*TEM is part of the processing software, it must remain running during processing. Protocols such as NX, VNC or RDP can be used to start *cis*TEM locally, then log off and connect again remotely.